# Jamaican fruit bats' competence for Ebola but not Marburg virus is driven by intrinsic differences

Sarah van Tol [1], Julia R. Port[1,2], Robert J. Fischer [1], Shane Gallogly[1], Trenton Bushmaker [1], Amanda Griffin[1], Jonathan E. Schulz[1], Aaron Carmody [3], Lara Myers [3], Daniel E. Crowley[4], Caylee A. Falvo [4], Jade C. Riopelle[1], Arthur Wickenhagen [1], Chad Clancy [5], Jamie Lovaglio [5], Carl Shaia [5], Greg Saturday [5], Jessica Prado-Smith [5], Yi He[6], Justin Lack[3], Craig Martens[3], Sarah L. Anzick[3], Lon V. Kendall[7], Tony Schountz [7], Raina K. Plowright[4], Andrea Marzi [1] & Vincent J. Munster [1] ✉

Ebola virus (EBOV) and Marburg virus (MARV) are zoonotic filoviruses that cause hemorrhagic fever in humans. Correlative data implicate bats as natural EBOV hosts, but neither a full-length genome nor an EBOV isolate has been found in any bats sampled. Here, we model filovirus infection in the Jamaican fruit bat (JFB), *Artibeus jamaicensis,* by inoculation with either EBOV or MARV through a combination of oral, intranasal, and subcutaneous routes. Infection with EBOV results in systemic virus replication and oral shedding of infectious virus. MARV replication is transient and does not shed. In vitro, JFB cells replicate EBOV more efficiently than MARV, and MARV infection induces innate antiviral responses that EBOV efficiently suppresses. Experiments using VSV pseudoparticles or replicating VSV expressing the EBOV or MARV glycoprotein demonstrate an advantage for EBOV entry and replication early, respectively, in JFB cells. Overall, this study describes filovirus species-specific phenotypes for both JFB and their cells.

Bats are the presumed reservoir hosts for viruses of five filovirus genera. Extensive experimental infection[1–5] and epidemiological evidence[6–8] supports that the Egyptian rousette bat (ERB) (*Rousettus aegyptiacus*) is the natural host of the orthomarburgviruses, Marburg virus (MARV) and Ravn virus (RAVV). ERBs cave roost in large groups and antibodies against orthomarburgviruses are consistently found in several ERB populations[6–11]. Infectious virus has been found mainly in the saliva[1–3], but virus is also detected in urine and feces at lower levels[3]. Biting, social grooming, contaminated food (saliva), or aerosolized urine or feces could facilitate transmission although the natural route(s) remain unconfirmed[12]. Dehong virus, the newest member of the filovirus genus *Delovirus*, was isolated from Leschenault's rousette (*R. leschenaultii*) lung samples in China[13]. A serological survey and tissue sampling of bats in Yunnan, China suggests Dehong virus circulates within the bat population. Lloviu virus (LLOV), a *Cuevavirus*, was isolated from European bats[14–18]. Initial detection of LLOV in Schreiber's bats (*Miniopterus schrebersii*) was linked with die-offs[15], but no causal link between LLOV infection and death was established.

[1]Laboratory of Virology, National Institute of Allergy and Infectious Diseases, National Institutes of Health, Hamilton, MT, USA. [2]Laboratory of Transmission Immunology, Helmholtz Centre for Infection Research, Braunschweig, Germany. [3]Research Technologies Branch, National Institute of Allergy and Infectious Diseases, National Institutes of Health, Hamilton, MT, USA. [4]Department of Public and Ecosystem Health, College of Veterinary Medicine, Cornell University, Ithaca, NY, USA. [5]Rocky Mountain Veterinary Branch, National Institute of Allergy and Infectious Diseases, National Institutes of Health, Hamilton, MT, USA. [6]Fermentation Facility, Biochemistry and Biophysics Center, National Institutes of Health, Bethesda, MD, USA. [7]Department of Microbiology, Immunology, and Pathology, Colorado State University, Fort Collins, CO, USA. ✉e-mail: vincent.munster@nih.gov

In addition to virus isolation, full genomes of other filoviruses have been detected in bats. The full-length genome of the *Dianlovirus* Měnglà virus (MLAV) was sequenced from *Rousettus spp.* in China[19]. In 2018, the orthoebolavirus Bombali virus (BOMV) was identified in Angolan (*Mops condylurus*) and little free-tailed (*Chaerephon pumilus*) bats in Sierra Leone[20]. Since BOMV's identification, molecular and serological data indicate an extensive distribution within Africa[21–25].

Aside from BOMV, the molecular evidence for orthoebolaviruses naturally circulating in bats is sparse[26]. Serological data support that several bat species, including the hammer-headed fruit bat (*Hypsignathus monstrosus*), are exposed to filoviruses[9,27]. Oppositionally, a study that evaluated viral molecular signatures of reservoir hosts predicted that Bundibugyo virus and Taï Forest virus circulate in a primate host[28]. The lack of molecular evidence for orthoeboloviruses in bats, aside from BOMV, suggests that either the incorrect bat species are being sampled or a non-bat host may be the natural host unlike the other bat-associated filoviruses[29].

Chiroptera, the mammalian order of bats, includes over 1400 species. Bats are diverse and occupy various niches and specialize in different diets and behaviors. The two bat suborders, Yinpterochiroptera and Yangochiroptera, diverged ~63 million years ago[30]. Filoviruses successfully infect bats from both suborders which suggests co-evolution between bats and filoviruses. The factors that predict which bat species could support replication and transmission of the different bat-associated filoviruses are unknown. For example, ERBs support disseminated replication and shedding of MARV, whereas infection with the various orthoebolaviruses is limited and does not uniformly induce an antibody response[2]. Additionally, the physiological and environmental stressors that regulate filovirus shedding intensity and transmission remain largely unknown. Due to gaps in knowledge regarding within host parameters that dictate filovirus-bat species compatibility, we evaluated the potential of Jamaican fruit bats (JFB) (*Artibeus jamaicensis*) to support Ebola virus (EBOV) and MARV replication.

In this work, we observe that JFBs support non-lethal, disseminated EBOV infection with minimal-to-mild clinical signs and histopathologic changes and shed infectious virus. In contrast, MARV replication is limited primarily to the inoculation site, no MARV RNA is detectable in swabs, and an antibody response is varaible. The difference between EBOV and MARV replication in live bats correlates with in vitro data that shows EBOV is more efficient than MARV in JFB cell entry and antagonism of the type I interferon (IFN-I) response. These differences between EBOV and MARV may aid in the identification of key filovirus-host interactions that are required for complementarity. Further, the JFB can be a useful EBOV infection model to study the physiological and environmental stressors that impact infection kinetics and viral shedding.

## Results

### Infection of JFBs with EBOV or MARV does not cause overt clinical disease

Bats were inoculated with EBOV strain Mayinga ($n = 12$) or MARV strain Ozolin ($n = 12$) via the oral, intranasal, and subcutaneous routes. MARV strain Ozolin was chosen since it was the closest relative to MARV bat 371, used for the ERB-MARV model, available to us at the time of the study. At 3- and 7-DPI, four EBOV- and four MARV-infected bats were euthanized, and the remaining animals were monitored through 28 DPI (Fig. 1A). The EBOV-infected bats developed mild hypothermia with a statistically significant decrease in body temperature at 4-, 8-, and 10-DPI relative to baseline temperatures (Fig. 1B). All EBOV-infected bats returned to baseline body temperatures by 28 DPI. There were no statistically significant changes in body weight compared to baseline in EBOV-infected bats (Fig. 1C). The MARV-infected bats did not have significant changes in body temperature (Fig. 1D) or weight

(Fig. 1E). MARV05 was pregnant and gained weight throughout the study.

Since lymphopenia is a hallmark of filovirus disease in pathogenic hosts[31,32], we collected whole blood at necropsy for complete blood cell counts. There were no significant changes in total white blood cells (WBC), lymphocytes, or neutrophils relative to baseline in either EBOV- or MARV-infected bats (Fig. 1F, G). In EBOV-infected bats, the number of monocytes increased significantly at 3- and 7-DPI (Fig. 1F). The number of circulating monocytes did not change after MARV infection (Fig. 1G). We did not observe any other statistically significant changes in hematology (Supplementary Table 1) or serum chemistries (Supplementary Table 2).

### JFBs shed infectious EBOV orally

Prior to infection, every 2 days through 14 DPI, 21 DPI, and at necropsies we collected oropharyngeal and rectal swabs to monitor viral shedding (Fig. 1A). EBOV RNA was detected in oropharyngeal swabs 2-10 DPI, peaking at 6 DPI (Fig. 2A), and rectal swabs 2-6 DPI (Fig. 2B). MARV RNA was not detectable in any oropharyngeal or rectal swabs (Fig. 2C, D). Infectious EBOV was present in oropharyngeal swabs, up to 2.5 $\log_{10}$ TCID$_{50}$/mL (Fig. 2E), but not in rectal swabs (Fig. 2F). No infectious MARV was detected in 6 DPI oropharyngeal swabs (Fig. 2G).

### EBOV, but not MARV, infection is disseminated in JFBs

Tissues collected on the sequential necropsy days were analyzed for the presence of viral RNA, infectious virus, and viral antigen. EBOV RNA was detected in all tissues evaluated, skin at the inoculation site (SIS) (8/8), salivary gland (5/8), lung (7/8), liver (8/8), spleen (7/8), kidney (5/8), and reproductive organs (3/8) (ovary for female, testis for male), for at least one bat at 3 and 7 DPI necropsies (Fig. 3A). The highest levels of EBOV RNA were detected in the SIS (5.5–7.5 $\log_{10}$ copies/g) at 3 and 7 DPI (Fig. 3A). EBOV RNA remained detectable in the spleens (4.5–5.9 $\log_{10}$ copies/g) of all four bats and one SIS sample (2.9 $\log_{10}$ copies/g) at 28 DPI while the other tissues were negative (Fig. 3A). Titration of the tissue samples largely reflected the RT-qPCR data with infectious virus present in at least one sample for all tissues tested (Fig. 3B). We were unable to evaluate the presence of infectious virus in spleen samples collected at 28 DPI due to insufficient sample. Minimal MARV RNA was detected in the SIS (8/8, 2.7–4.7 $\log_{10}$ copies/g), liver (1/8, 2.7 $\log_{10}$ copies/g), and spleen (2/8, 3.5–4.6 $\log_{10}$ copies/g) at 3 and 7 DPI, and all other samples were negative (Fig. 3C). Minimal infectious MARV was detected in SIS and liver samples collected at 3- and 7 DPI (Fig. 3D). All serum samples were negative for viral RNA, which is likely a result of missing a narrow viremia window (Fig. 3E, F).

Injection site skin, salivary gland, cervical lymph node, soft palate/tonsil, lung, heart, thymus, liver, spleen, kidney, adrenal gland, urinary bladder, gonad, reproductive tract, gastrointestinal tract and eye were evaluated by histopathologic analysis for evidence of viral induced inflammation (Supplementary Fig. 1). In EBOV-infected bats, sections of SIS revealed granulomatous inflammation in the subcutis. Subcuticular inflammation was most pronounced in bats 03, 05, 09, and 10, reaching moderate inflammation. When present for evaluation, neutrophilic lymphadenitis was observed in cervical lymph nodes ranging from minimal to moderate severity. Sporadic splenic changes were observed at all timepoints in EBOV-infected bats including minimal to moderate splenic lymphoid depletion ($n = 8$) and minimal to mild lymphoid follicular necrosis with neutrophilic influx ($n = 5$). In MARV-infected bats, minimal to mild subcutaneous perivascular cuffing, pyogranulomatous panniculitis, and focal myocyte regeneration were observed sporadically at all evaluated timepoints. No significant histopathologic lesions were observed in evaluated sections of tonsil/palate, salivary gland, kidney, adrenal gland, lung, heart, thymus, liver, urinary bladder, gastrointestinal tract or eye from either EBOV- or MARV-infected JFBs.

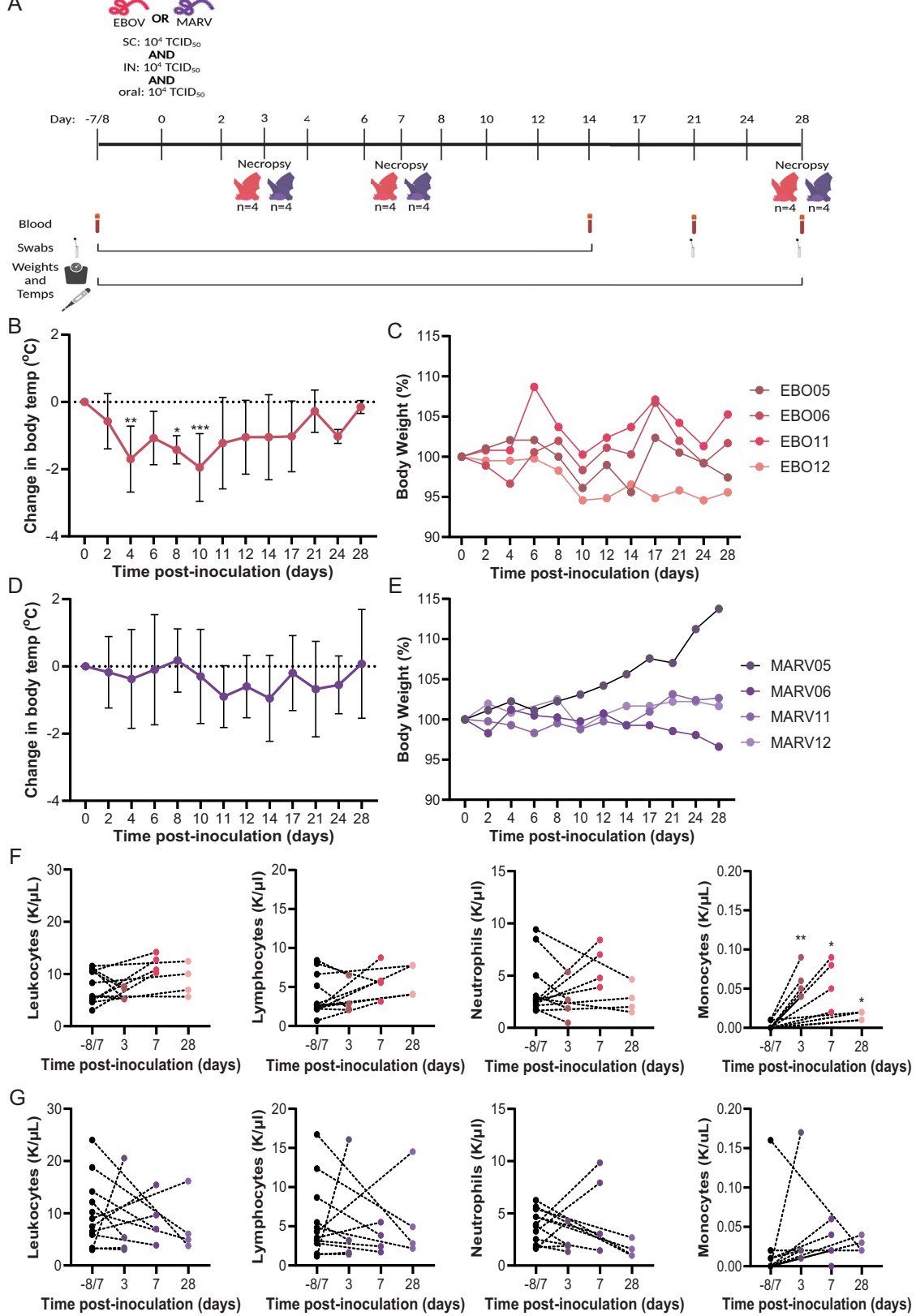

## EBOV infection induces robust innate antiviral and humoral responses

Host gene expression analysis using RT-qPCR was performed on SIS, lung, liver, and spleen samples collected at 3- and 7 DPI necropsies for both interferon stimulated genes (ISGs) and inflammatory cytokines. In the SIS, expression of ISGs, *Isg15*, *Ifit1*, *Mx1*, *Oas1*, and *Ddx58*, was

induced significantly in EBOV-infected bats at 3 DPI and remained marginally elevated at 7 DPI (Fig. 4A). EBOV infection significantly induced *Il6* expression in the SIS at 7 DPI compared to healthy control bats (Fig. 4B). In MARV-infected bats, *Ifit1* at 3 DPI was induced and *Il1b* mRNA expression was inhibited at 3 and 7 DPI compared to the healthy controls (Fig. 4A, B). At 3 DPI with EBOV, all ISG mRNAs, except RIG-I,

**Fig. 1 | Infection of JFB with EBOV or MARV does not cause overt clinical disease. A** Depiction of challenge study of Jamaican fruit bats with EBOV-Mayinga (magenta) or MARV-Ozolin (purple) via subcutaneous (SC), intranasal (IN), and oral routes. Image created in BioRender [https://BioRender.com/a66b266]. **B** Change in body temperature of bats ($n = 4$) infected with EBOV followed through day 28. Two-way ANOVA with Dunnett's multiple test comparison. Data plotted as mean ± S.D. $P$ values for comparisons to baseline weight are indicated. *** <0.001, ** <0.01, * <0.05. **C** Percent body weight of bats ($n = 4$) infected with EBOV followed through day 28. Two-way ANOVA with Dunnett's multiple test comparison. **D** Change in body temperature of bats ($n = 4$) infected with MARV followed through day 28. Two-way ANOVA with Dunnett's multiple test comparison. Data plotted as

mean ± S.D. **E** Percent body weight of bats ($n = 4$) infected with MARV followed through day 28. Two-way ANOVA with Dunnett's multiple test comparison. **F** Number of circulating white blood cells (WBC), lymphocytes, neutrophils, and monocytes in EBOV-infected bats ($n = 4$) at baseline and at necropsies. A mixed-effects model with matching and Dunnett's multiple comparison test was applied to evaluate change in each cell population at necropsy compared to the matched baseline value. **G** Number of circulating WBC, lymphocytes, neutrophils, and monocytes in MARV challenged bats ($n = 4$) at baseline and at necropsies. A mixed-effects model with matching and Dunnett's multiple comparison test was applied to evaluate change in each cell population at necropsies compared to the matched baseline value.

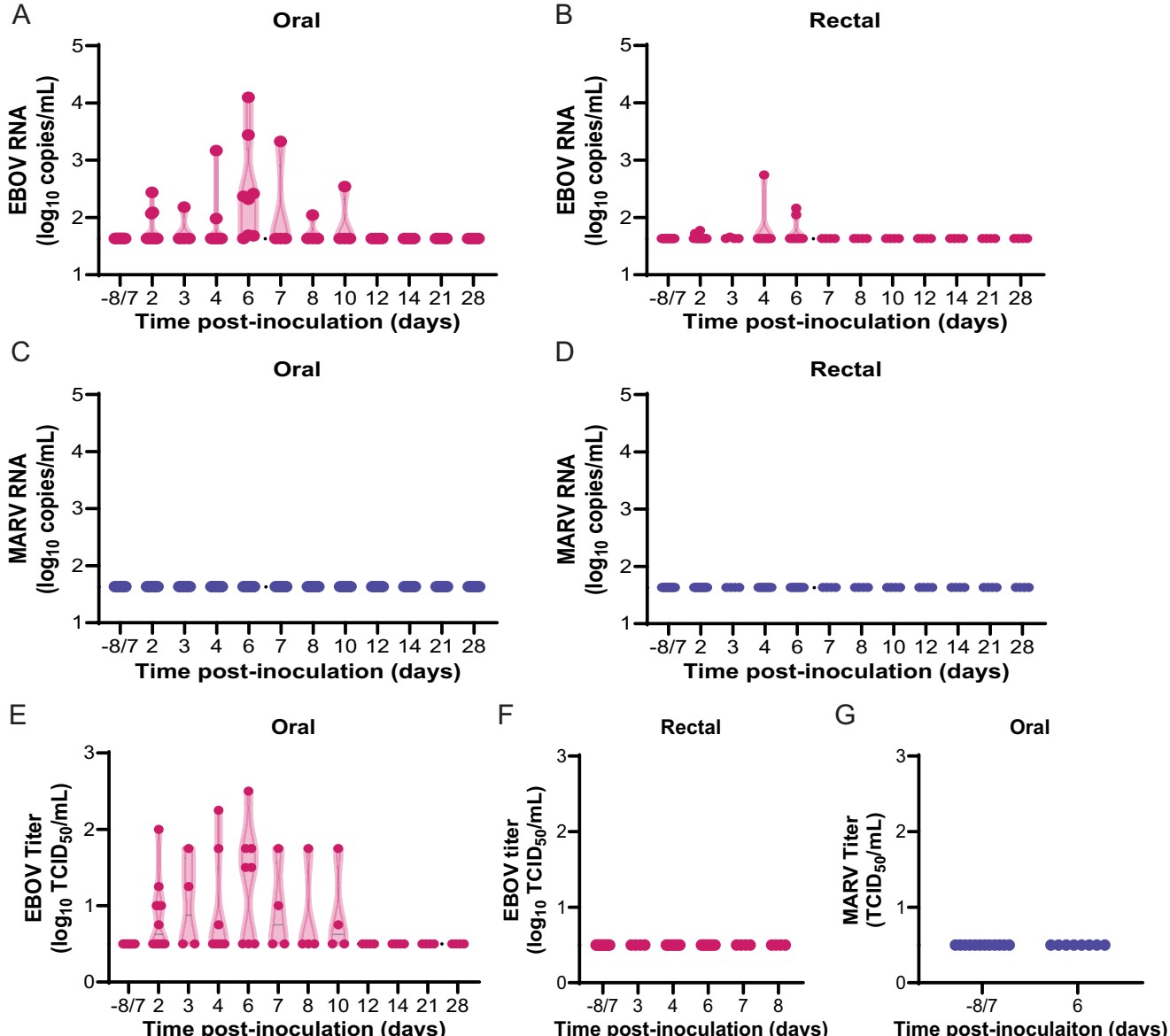

**Fig. 2 | Infected JFBs shed infectious EBOV orally.** RT-qPCR for EBOV (**A**, **B**) or MARV (**C**, **D**) RNA in the oral (**A**, **C**) or rectal (**B**, **D**) swabs. Limit of detection 1.63 $\log_{10}$ copies/mL. **E** Infectious titer of EBOV in the oral swabs. Limit of detection 0.5 $\log_{10}$ TCID$_{50}$/mL. **F** Infectious titer of EBOV in the rectal swabs in span of days

where viral RNA was detected in swabs. Limit of detection 0.5 $\log_{10}$ TCID$_{50}$/mL. **G** Infectious titer of MARV in the oral swabs at baseline and on day 6 post-challenge. Limit of detection 0.5 $\log_{10}$ TCID$_{50}$/mL.

were significantly induced in the liver (Fig. 4C). Expression of *Il1b* (7 DPI) and *Tnfa* (3 and 7 DPI) were significantly induced in the liver of EBOV-infected bats (Fig. 4D). *Ifit1* (3 DPI) and *Tnfa* (3- and 7 DPI) were elevated significantly in the liver of MARV-infected bats (Fig. 4C, D). In the 3 DPI spleens, all ISG mRNAs were induced in EBOV-infected bats

and *Ifit1* was induced in MARV-infected bats (Fig. 4E). *Tnfa* was significantly induced in the spleen of EBOV- and MARV-infected bats 7 DPI, and *Il6* mRNA was elevated significantly at 3 and 7 DPI after EBOV infection (Fig. 4F). In the lung *Isg15*, *Ifit1*, and *Mx1* were the only ISGs with significantly increased transcription in EBOV-infected bats, and

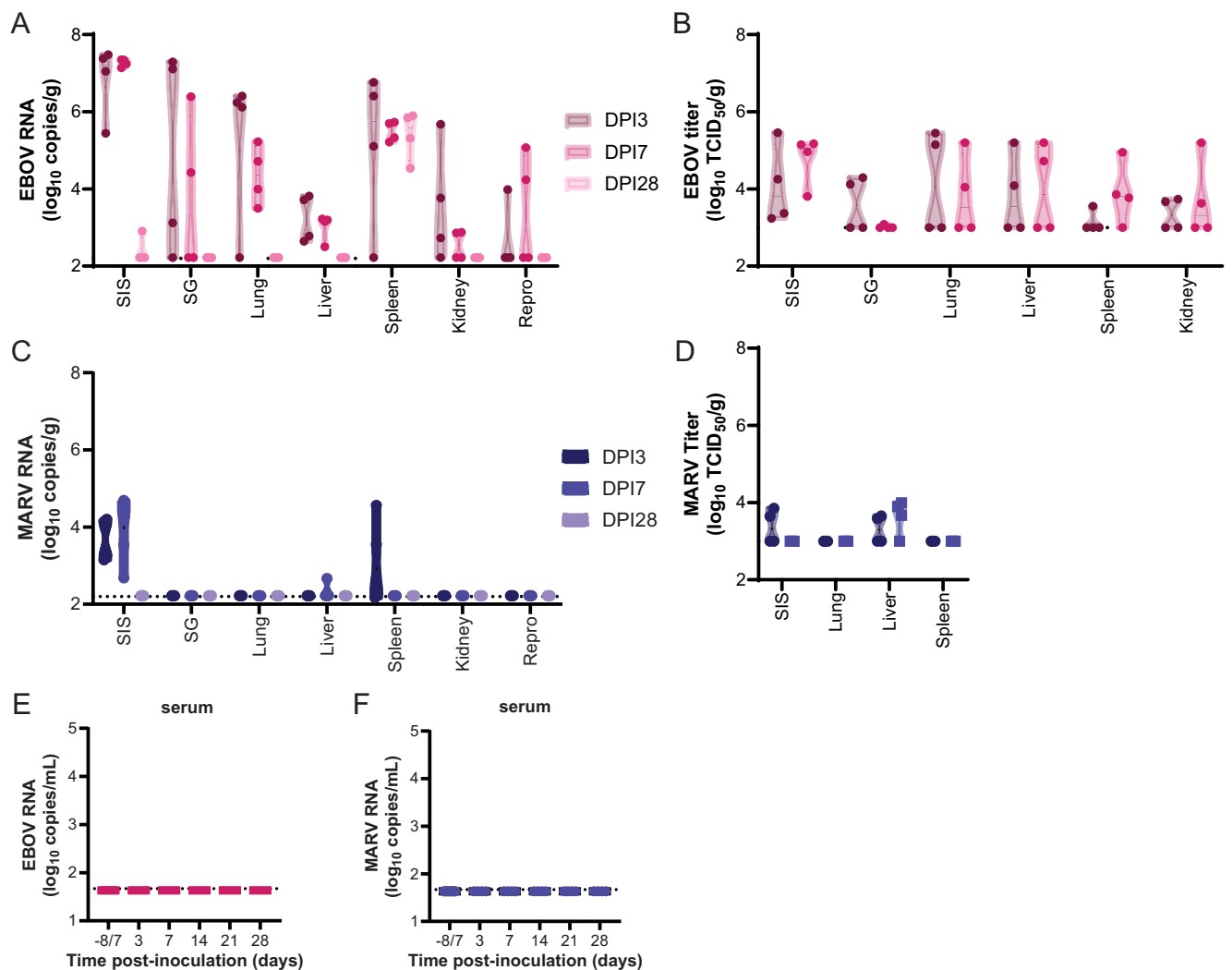

**Fig. 3 | EBOV but not MARV infection is disseminated in JFBs. A** RT-qPCR for EBOV in tissue samples collected at 3-, 7-, and 28 days post-infection (DPI) ($n = 4$). Limit of detection 2.2 $\log_{10}$ copies/g. **B** Infectious titer of EBOV in tissues collected at D3, 7, and 28 ($n = 4$). Limit of detection 3.0 $\log_{10}$ copies/g. **C** RT-qPCR for MARV in tissue samples collected at necropsy day (DPI) 3, 7, and 28 ($n = 4$). Limit of detection 2.2 $\log_{10}$ copies/g. **D** Infectious titer of MARV in tissues collected at D3, 7, and 28 ($n = 4$). Limit of detection 3.0 $\log_{10}$ copies/g.RT-qPCR. EBOV (**E**) or MARV (**F**) in serum samples collected at baseline, 3-, 7-, 14-, 21-, and 28 days post-infection (DPI). Limit of detection 1.67 $\log_{10}$ copies/mL.

ISG expression in MARV-infected bats did not differ from healthy controls (Supplementary Fig. 2A). Proinflammatory cytokine genes were not differentially expressed in either EBOV- or MARV- infected bats at either timepoint (Supplementary Fig. 2B). Of note, the magnitude of induction for ISGs was higher than for pro-inflammatory cytokines in all tissues.

We performed transcriptomic analysis on 3 DPI liver samples, because we observed viral RNA and infectious virus for both EBOV- and MARV-infected bats. Comparing EBOV-infected bats with healthy controls, we found 241 upregulated and 460 downregulated genes; comparing MARV-infected bats with healthy controls, we found 230 upregulated and 516 downregulated genes (significance level = 0.05, foldchange > 2). The Gene Ontology enrichment analysis revealed that nearly all the top ten pathways enriched in EBOV-infected bats were also enriched in those infected with MARV and are involved in the regulation of the immune response (Supplementary Table 3). When comparing EBOV- and MARV-infected bats, we observed 98 upregulated and 146 downregulated genes in EBOV- relative to MARV-infected bats (significance level = 0.05, foldchange > 2). The pathways differentially regulated between the two filoviruses are involved in immune

regulation (Supplementary Table 4). Interestingly, some genes involved in antigen processing and presentation are induced in EBOV- but not MARV-infected bats. Since the amount of viral RNA in the liver is higher in EBOV- than MARV-infected bats, the stronger of induction of genes involved in immune regulation as well as antigen processing and presentation is not unexpected.

GP-specific antibodies were measured using ELISA on serum collected from the bats monitored through 28 DPI. All four of the EBOV-infected bats seroconverted with EBOV GP-specific binding antibodies steadily increasing after infection (Fig. 4G). The titers of EBOV neutralizing antibodies at 28 DPI were 20 for EBO06 and 40 for the other bats (Fig. 4H). Only one of the four MARV-infected bats, MARV80, seroconverted (Fig. 4I). MARV80 was the only MARV-infected bat with a detectable MARV-neutralizing titer (20) (Fig. 4J).

At the 28 DPI necropsies, we cryopreserved spleens from EBOV-infected bats to evaluate lymphocyte populations compared to healthy controls with flow cytometry. The proportion of total T (CD3+) and B (CD79a+) lymphocytes in EBOV-infected bats ($n = 4$) did not differ significantly from the healthy controls ($n = 8$), although the proportion of T lymphocytes was elevated in spleens of bats infected with EBOV

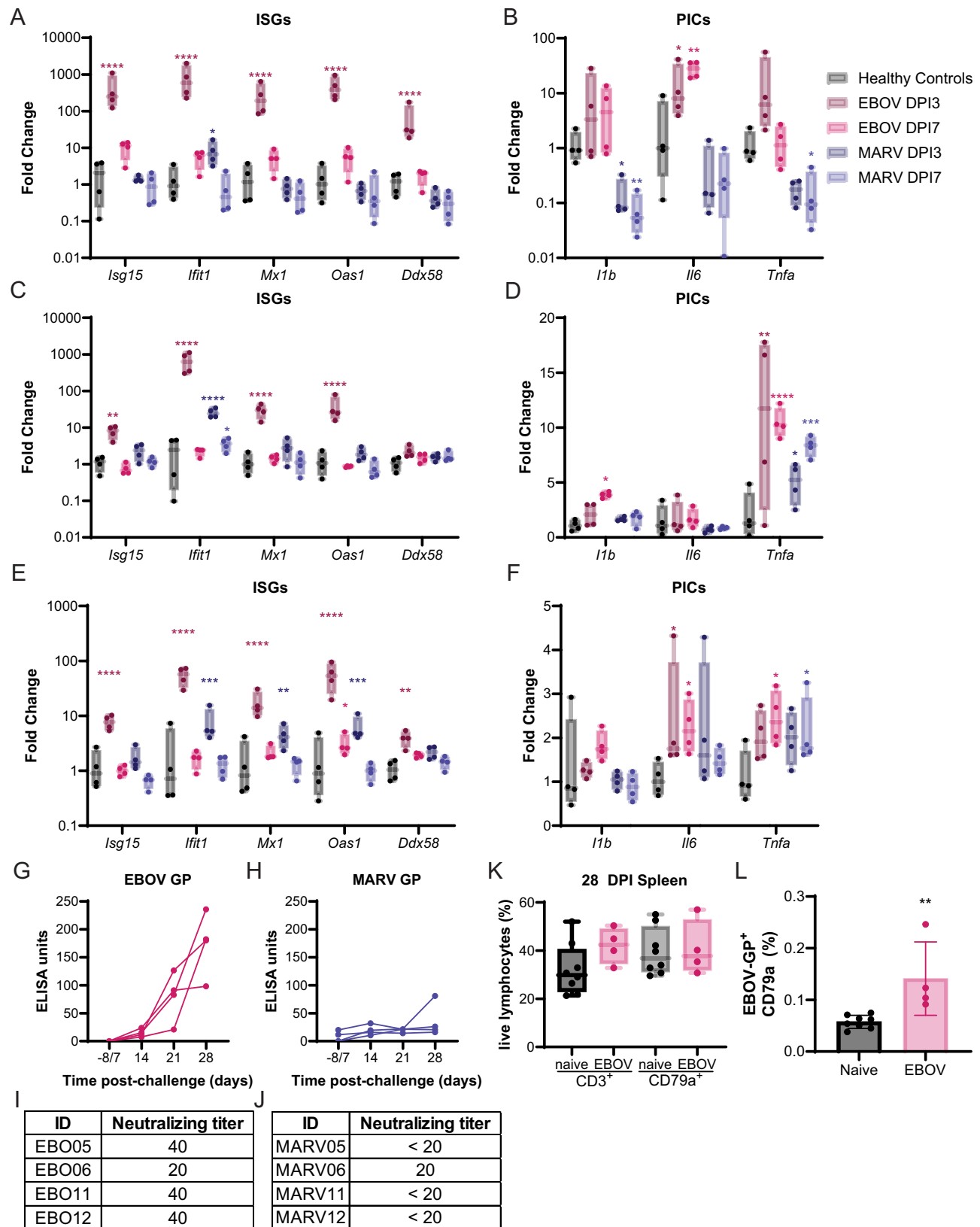

(Fig. 4K). EBOV-infected bats had a distinct EBOV GP binding B cell population (mean 0.15%) compared to the healthy controls (Fig. 4L).

Early B cell responses are characterized by the proliferation of plasmablasts and the formation of germinal centers in lymphoid tissues. Either process can lead to an expansion of specific B cell clonal lineages. Therefore, we analyzed the size of B cell clones in healthy

controls ($n = 10$) and EBOV-infected bats ($n = 3$) at 3-, 7-, and 28 DPI. Because MARV-infected bats did not mount a MARV GP-specific humoral response we did not evaluate their B cell clonal lineages. Among EBOV infected bats, the median count of B cell clones increased by 3 DPI and remained elevated until 7 DPI. The median count of B cell clones continued to increase from pre-infection levels

**Fig. 4 | EBOV infection induces robust innate antiviral and adaptive immune responses.** RT-qPCR of interferon stimulated gene (ISG) (**A**, **C**, **E**) or pro-inflammatory cytokine (PIC) (**B**, **D**, **F**) mRNA in the skin at inoculation site (**A**, **B**), liver (**C**, **D**), or spleen (**E**, **F**) collected from healthy control bats (n = 4) and at day (**D**) 3 or 7 necropsy of EBOV or MARV-infected bats (n = 4). **A**–**F** $\Delta C_T$ values were normalized to the average $\Delta C_T$ value of the healthy control bats to calculate $\Delta\Delta C_T$. Fold change of each gene at necropsy day was compared to the healthy controls using a two-way ANOVA with Dunnett's multiple test comparison. **G** Serum binding antibodies and (**H**) neutralizing titers were determined via ELISA assay with a standard curve of EBOV glycoprotein (GP) or neutralization assay, respectively. Neutralizing titer is presented as the lowest dilution that protected 50% of wells in a $TCID_{50}$ assay. **I** Serum binding antibodies and (**J**) neutralizing titers were determined via ELISA assay with a standard curve of MARV GP or neutralization assay, respectively. Neutralizing titer is presented as the lowest dilution that protected 50% of wells in a $TCID_{50}$ assay. **K** Cryopreserved splenocytes from naïve controls (n = 8) and D28 EBOV (n = 4) infected bats were analyzed for intracellular expression of CD3ε and CD79a to identify the proportion of T cells and B cells from a live lymphocytes gate. **L** CD79a+ B cells from cryopreserved splenocytes, naïve controls (n = 8), and D28 EBOV (n = 4) infected bats, were analyzed for interaction with Alexa-fluor 598 conjugated recombinant EBOV-GP receptor binding domain. The percentage of live CD79a$^+$ B cells with a positive EBOV-GP staining signal after subtracting the fluorescence minus one control signal. Data plotted as mean ± S.D. A two-sided Mann-Whitney statistical test was used to assess statistical significance. **A**–**F**, **K** Box defines the upper (75th percentile) and lower (25th percentile) quartiles with whiskers extending from minimum to maximum with all values shown, and the line as the median. *P* values adjusted for multiple comparisons <0.05 for comparisons versus healthy controls are indicated. **** <0.0001, *** <0.001, ** <0.01, * <0.05.

to 28 DPI (Supplementary Fig. 3A). However, due to significant individual-level variation at all time points, it was likely underpowered to detect statistical significance. This pattern was primarily observed among IgG sequences, and no clear pattern was observed among IgM sequences (Supplementary Fig. 3B).

Next, we analyzed the frequency of VH and J gene segments. We restricted our analysis to only the most common VH gene family, the mouse homolog V5. Among IgG BCR sequences, the frequency of V5 sequences visually appeared to decrease at 3 DPI and return to pre-infection frequencies at 28 DPI (Supplementary Fig. 3C). We analyzed the proportion of J segments focusing specifically on the most common J segment sequence, the mouse IGHJ3 gene homolog. Among IgM BCR sequences, the frequency of the J3 gene segment showed a statistically significant increase at 3 DPI (Supplementary Fig. 3D). The J3 segment frequencies had returned to the level of healthy controls by 28 DPI.

### EBOV enters and replicates more efficiently than MARV in JFB cells

We previously published a study showing that JFB cells support the replication of both EBOV and MARV in vitro, and the growth of MARV seemed delayed compared to EBOV[33]. To determine if MARV growth is attenuated in JFB cells, we titrated the stocks of three EBOV strains (Kikwit, Makona C05, and Mayinga), and three MARV strains (Angola, Musoke, Ozolin), on Vero E6 and primary JFB uropatagium-derived fibroblasts (AjUFi_RML6). The titers of all the EBOV strains on JFB cells (Kikwit: 6.0 $\log_{10}$ $TCID_{50}$/mL, Makona: 6.3 $\log_{10}$ $TCID_{50}$/mL, Mayinga: 6.9 $\log_{10}$ $TCID_{50}$/mL) were within 20-fold of the titers on Vero E6 cells (Kikwit: 6.6 $\log_{10}$ $TCID_{50}$/mL, Makona: 7.6 $\log_{10}$ $TCID_{50}$/mL, Mayinga: 7.5 $\log_{10}$ $TCID_{50}$/mL), whereas the MARV titers were 2000–32,622 fold lower on AjUFi_RML6 cells (Angola: 3.4 $\log_{10}$ $TCID_{50}$/mL, Musoke: 2.2 $\log_{10}$ $TCID_{50}$/mL, Ozolin: 3.6 $\log_{10}$ $TCID_{50}$/mL) than on Vero E6 cells (Angola: 6.7 $\log_{10}$ $TCID_{50}$/mL, Musoke: 6.7 $\log_{10}$ $TCID_{50}$/mL, Ozolin: 7.1 $\log_{10}$ $TCID_{50}$/mL) (Fig. 5A).

To address this innate difference of viral growth on JFB cells, the contribution of viral entry was assessed. Non-replicating VSVΔGFP particles pseudotyped with either EBOV-Mayinga or MARV-Musoke GP, were titrated on Vero E6, AjKi_RML1, AjKi_RML2, and RoNi cells. The ratio of the VSVΔGFP-EBOV and VSVΔGFP-MARV titers were determined for each cell type and normalized to the Vero E6 ratio. For both JFB cell lines, VSVΔGFP-EBOV entered more efficiently than the VSVΔGFP-MARV (AjKi_RML1: 3.5-fold, AjKi_RML2: 5.8-fold) (Fig. 5B). Although the VSVΔGFP-MARV trended toward more efficient entry on ERB cells, the difference was not significant (RoNi: 0.6-fold) (Fig. 5B). To assess how the difference in entry may impact viral replication, we titrated VSVwild-type-GFP (VSVwt-GFP), VSV-EBOVmayinga-GFP (VSV-EBOV-GFP), or VSV-MARVozolin-GFP (VSV-MARV-GFP) on Vero E6 or AjKi_RML2 cells. Unlike WT EBOV and MARV, all three VSVs grew to comparable titers on both cell lines (Fig. 5C). We also evaluated the replication kinetics of these VSVs on AjKi_RML2 cells (Supplementary

Fig. 4A) and Vero E6 cells (Supplementary Fig. 4B). After normalization of the infection ratio of VSV-EBOV-GFP and VSV-MARV-GFP on AjKi_RML2 cells to the ratio on Vero E6 cells to account for intrinsic differences between the viruses' replication, VSV-EBOV-GP grew to a titer 13.6-fold higher than VSV-MARV-GP at 16 HPI on JFB cells (Fig. 5D). At 48- and 72 HPI the infection ratio of the two viruses was within 2-fold.

### EBOV antagonizes JFB IFN-I signaling more efficiently than MARV

JFB and ERB cells were infected with EBOV-Mayinga or MARV-Ozolin to monitor replication kinetics and activation of the innate antiviral response. Consistent with previous results, MARV replication lagged from 24 (200-fold) through 72 (25-fold) HPI compared to EBOV in AjKi_RML2 cells, but both viruses reached similar titers at 96 (3-fold) and 120 (1.6-fold) HPI (Fig. 6A). Although EBOV replicated to a higher titer compared to MARV at 72 HPI, MARV infection induced phosphorylation of STAT1 and STAT2 and expression of ISGs RIG-I and STAT1 as early as 72 HPI, whereas EBOV infection did not detectably activate the IFN-I signaling pathway through 120 HPI (Fig. 6B). MARV infection induced expression of *Ifnb1* (Fig. 6C) and ISG (Fig. 6D) mRNAs 100- to 32,000-fold beginning at 72 HPI, and the same genes remained at mock levels in EBOV-infected cells. *Tnfa* expression was induced earlier in MARV-infected cells, but by 120 HPI levels were similar in both EBOV and MARV-infected cells (Fig. 6D). Infection with either virus did not alter *Il6* expression more than 2-fold (Fig. 6E). In ERB cells, EBOV and MARV replication kinetics were identical (Supplementary Fig. 5A). MARV (48 HPI) infection induced *Ifnb* and ISG mRNAs earlier than EBOV (72 HPI) (Supplementary Fig. 5B, C). RoNi cells infected with either virus increased expression of *Ifnb1* and ISG mRNAs to similar peak levels. Similar to the JFB cells, neither filovirus induced *Il6* but both induced *Tnfa* similarly in ERB cells (Supplementary Fig. 5D). Despite similar peak levels of *Ifnb1* and ISG mRNAs, immunoblotting showed stronger activation of the IFN-I signaling pathway in MARV-infected cells (Supplementary Fig. 5E).

To ensure that the intrinsic differences in replication and IFN-I signaling were not strain-specific, multiple JFB cell lines were infected with three different strains of EBOV and MARV. Importantly, back titration showed that the inoculation dose was within 5-fold for all the strains (Supplementary Fig. 5F). All three EBOV strains replicated to higher titers than all three MARV strains at 24 and 72 HPI on AjKi_RML2 cells (Fig. 6F). At 72 HPI, only MARV-Ozolin induced expression *Ifnb1* and ISG mRNAs, but all three MARV strains induced these genes at 120 HPI (Fig. 6G, H). None of the three EBOV strains induced expression of the evaluated IFN-I pathway genes. EBOV-Kikwit induced *Tnfa* at 72 HPI, and infection increased *Tnfa* 3-32 fold at 120 HPI regardless of strain (Fig. 6I). Although the magnitude of the difference varied across cell lines, all EBOV strains replicated to higher titer than MARV strains on three additional JFB cell lines (AJi, AjKi_RML2, and AjLu_RML3) (Fig. 6J). Because we observed similar patterns in host gene expression

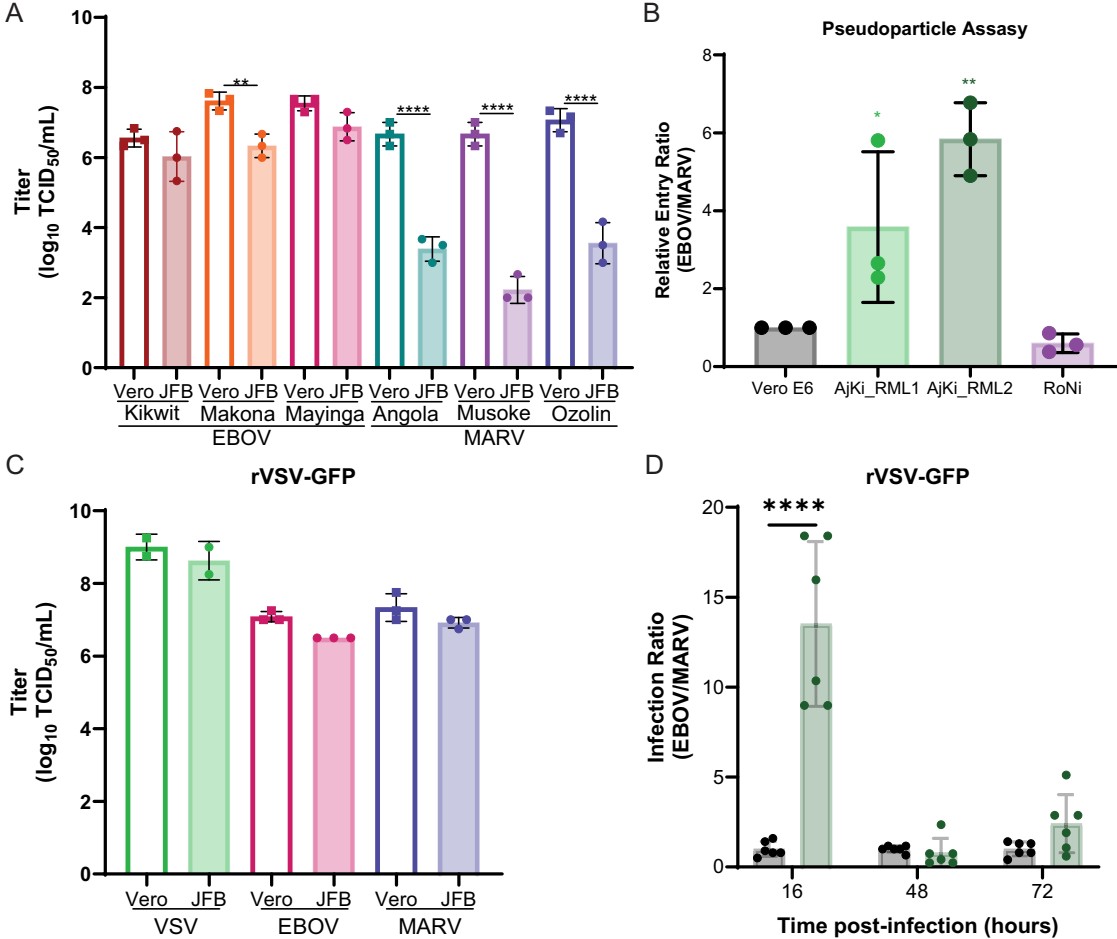

**Fig. 5 | EBOV enters and replicates in JFB cells more efficiently than MARV.**
**A** Infectious titers of three EBOV strains (Kikwit, Makona C05, and Mayinga) and MARV strains (Angola, Musoke, and Ozolin) stocks ($n = 3$) measured on Vero E6 and JFB uropatagium-derived fibroblasts (AjUFi_RML6). Three independent stock vials were titrated on different days. The titer of each virus was compared on the two cell lines using a two-way ANOVA with Dunnett's multiple test comparison.
**B** Replication incompetent vesicular stomatitis virus (VSV) with glycoprotein gene replaced with green fluorescent protein pseudotyped with either EBOV or MARV glycoprotein (GP) were titrated on Vero E6, Jamaican fruit bat kidney (AjKi_RML1 and AjKi_RML2), or Egyptian rousette bat kidney (RoNi) cells. To determine the relative entry ratio, we normalized the ratio of EBOV/MARV infectious units for each cell line and divided by the ratio on Vero E6 cells. Three independent experiments were performed with four technical replicates each experiment. One-way ANOVA with Dunnett's multiple test comparison. **C** Infectious titers of recombinant VSV

expressing GFP (rVSVwt-GFP) ($n = 2$) or GFP plus EBOV-Mayinga ($n = 3$) or MARV-Ozolin ($n = 3$) GP were measured on Vero E6 and AjKi_RML2. Two (VSVwt) or three (EBOV and MARV GP) independent stock vials were titrated on two different passages of cells. The titer of each virus was compared on the two cell lines using a two-way ANOVA with Dunnett's multiple test comparison. **D** Vero E6 and AjKi_RML2 cells were infected at multiplicity of infection (MOI) 0.005 for 1 h and supernatants were collected at 16-, 48-, and 72 h post-infection. The ratio of infectious titers for rVSV-, EBOV-GFP, or rVSV-MARV-GFP was calculated for both cell lines and normalized to the ratio on Vero E6 cells (Replication kinetics shown in Supplementary Fig. 4). Two independent experiments were performed with three technical replicates each experiment. The infection ratio at each time point was compared using a two-way ANOVA with Dunnett's multiple test comparison. **A**–**D** Data plotted as mean ± S.D. *P* values for comparisons to healthy controls are indicated. **** <0.0001, ** <0.01, * <0.05.

across the different virus strains on AjKi_RML2 cells and similar replication kinetics across all JFB cell lines, changes to host mRNAs were measured in the other JFB cell lines 120 HPI with EBOV-Mayinga and MARV-Ozolin. *Ifnb1* was induced in AjKi_RML1 and AjLu_RML3 cells infected with MARV but not EBOV (Fig. 6K). In AjKi_RML1 cells expression of all ISGs increased in MARV-infected cells compared with mock while EBOV infection did not alter ISG mRNA levels (Fig. 6L). ISG mRNAs increased in both EBOV and MARV infected AjLu_RML3 cells, but the magnitude of expression was higher in MARV-infected cells than EBOV-infected cells despite replication to lower titer throughout infection (Fig. 6J, L). AJi cells express very low levels of endogenous STAT1 compared to the other cell lines, which causes AJi cells to have a higher threshold for activation of the IFN-I signaling pathway. In AJi cells, levels of *Ifnb1* and ISG mRNAs did not change in either EBOV- or MARV-infected cells relative to mock (Fig. 6K, L). TNF mRNA levels did not differ in any of the cell lines infected with either virus (Fig. 6M).

Replication kinetics with all the EBOV and MARV strains were conducted on human and ERB cell lines to confirm that the differences were not virus-intrinsic. Overall, the titers of the EBOV and MARV strains were similar throughout the time course for human (Huh7) (Supplementary Fig. 5G) and two ERB kidney (RoNi and RASKM) cell lines (Supplementary Fig. 5H). In contrast to what is observed on JFB cells, all three MARV strains grew to a higher titer on ERB lung cells (RaLu) than all three EBOV strains at 120 HPI (Supplementary Fig. 5H). *Ifnb1* (Supplementary Fig. 5I), ISGs (Supplementary Fig. 5J), and *Tnfa* (Supplementary Fig. 5K) mRNAs were elevated in RoNi cells infected with each of the viruses at 120 HPI.

**MARV's inability to block IFN-I signaling attenuates replication**
Due to the activation of the IFN-I pathway in MARV-, but not EBOV-infected cells, the capacity of the viruses to antagonize IFN-I signaling was compared. MARV VP40 and EBOV VP24 block the activation of the

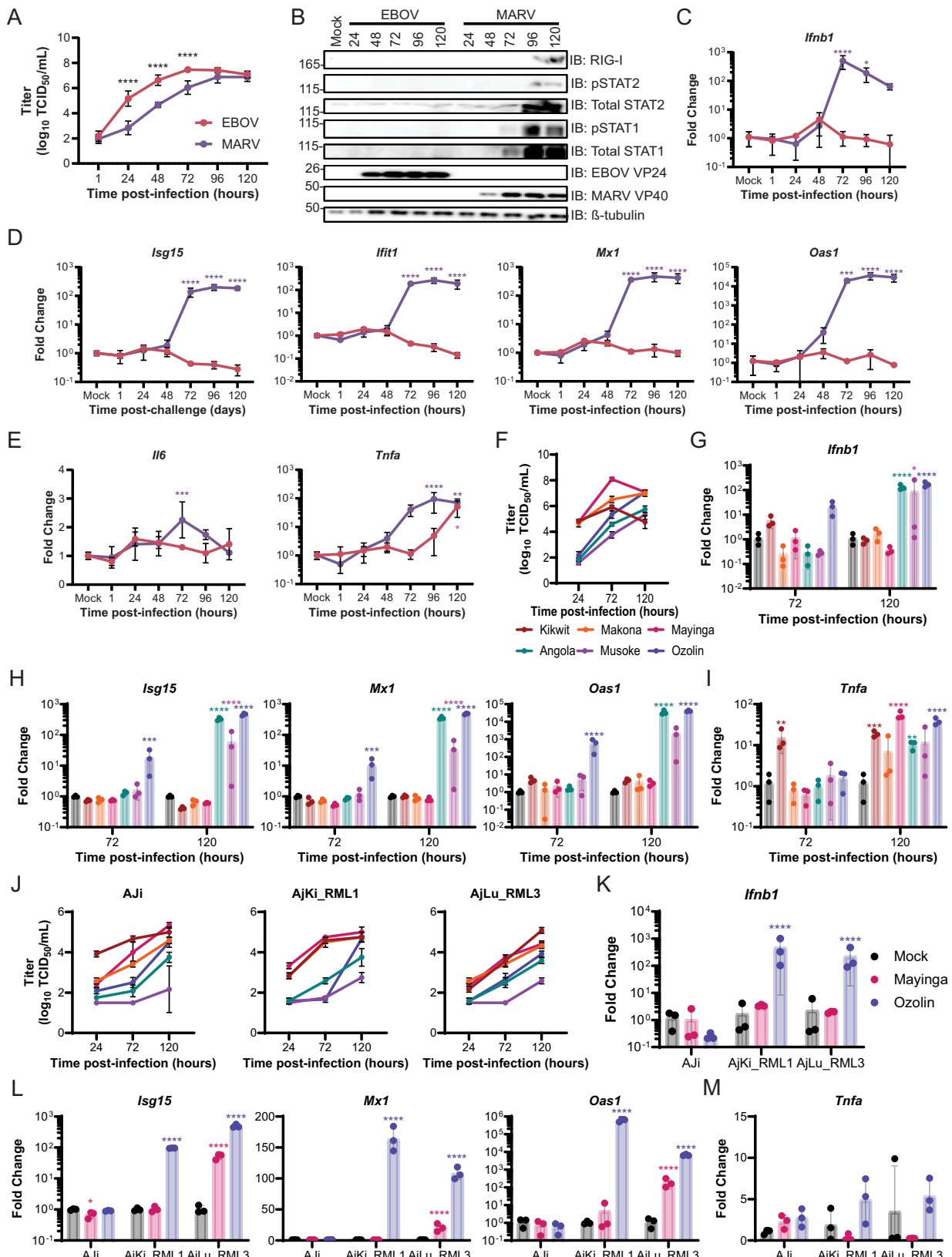

IFN-I response pathway at different points. MARV VP40 prevents the JAK1 phosphorylation and downstream phosphorylation of STAT1[34,35], whereas EBOV VP24 interacts with the NPI-1 subfamily of karyopherin alpha proteins to preclude pSTAT1's nuclear translocation[36–40] (Fig. 7A). To measure the functionality of EBOV and MARV IFN-I signaling antagonists, Huh7 cells were infected with EBOV-Mayinga or

MARV-Ozolin at MOI 1.5 for 24 h. Cells were stimulated with 100 ng/mL of recombinant Chiroptera IFN-β for 30 min prior to cytoplasmic-nuclear fractionation. As expected, pSTAT1 was present in both fractions of mock infected cells treated with IFN-β (Supplementary Fig. 6A). Cytoplasmic pSTAT1 levels were similar in mock and EBOV-infected cells treated with IFN-β, but nuclear pSTAT1 was 50% lower in

**Fig. 6 | EBOV antagonizes JFB's type I interferon pathway more efficiently than MARV. A** Infectious titers of EBOV-Mayinga and MARV-Ozolin on Jamaican fruit bat kidney cells (AjKi_RML2) infected with multiplicity of infection (MOI) 0.1. Data presented as $\log_{10}$ transformed values. Data are from two independent experiments with three technical replicates each. Two-way ANOVA with Sidak's multiple test comparison to evaluate the difference between EBOV and MARV titer at each time point. **B** Immunoblot of AjKi_RML2 cells infected with EBOV or MARV at MOI 0.1. The presented panel is representative of three independent immunoblots. RT-qPCR of *IFNb1* (**C**), interferon stimulated gene (**D**), or pro-inflammatory cytokine (**E**) mRNA in AjKi_RML2 cells infected with EBOV or MARV at MOI 0.1. **F** Infectious titers of three EBOV and MARV strains on AjKi_RML2 cells infected with MOI 0.1. Data presented as $\log_{10}$ transformed values. Data are from one experiment conducted in triplicate. RT-qPCR of *Infb1* (**G**), interferon stimulated gene (**H**), or

pro-inflammatory cytokine (**I**) mRNA in AjKi_RML2 cells infected with three different EBOV or MARV strains at MOI 0.1. **J** Infectious titers of three EBOV and MARV strains on Jamaican fruit bat immortalized kidney (AJi), primary kidney (AjKi_RML1), primary lung (AjLu_RML3) infected with MOI 0.1. Data presented as $\log_{10}$ transformed values. Data are from one experiment conducted in triplicate. RT-qPCR of *Ifnb1* (**K**), interferon stimulated gene (**L**), or *Tnfa* (**M**) mRNA in Jamaican fruit bat cells infected with three EBOV Mayinga or MARV Ozolin strains at MOI 0.1. **C–E**; **G–I**; **K–M** $\Delta C_T$ values were normalized to the average $\Delta C_T$ value of mock infected cells to calculate $\Delta\Delta C_T$. Data from three biological replicates. Fold change of each gene at each time point for EBOV and MARV-infected cells was compared to mock using a two-way ANOVA with Dunnett's multiple test comparison. **A, C–M** Data plotted as mean ± S.D. **A–E, G–I, K, L** *P* values for comparisons to mock cells are indicated. **** <0.0001, *** <0.001, ** <0.01, * <0.05.

---

EBOV-infected cells (Supplementary Fig. 6A, B). Both cytoplasmic and nuclear pSTAT1 was reduced over 90% in MARV-infected cells compared to mock (Supplementary Fig. 6A, B). Huh7 cells cannot produce IFN-β, thus the experiment does not require an IFN-I induction blockade after infection, and the cells were initially used to characterize MARV VP40 mediated IFN-I signaling antagonism[34]. Because the experiment requires a high amounts of virus replication to ensure expression of viral protein and IFN-I cannot be induced prior to the addition of exogenous IFN-β, we confirmed that the TBK1 and IKKε specific inhibitor BAY-985[41] blocked IFN-I induction in AjKi_RML2 cells after transfection with 500 ng/mL of high molecular weight poly (I:C) (Supplementary Fig. 7A, B). AjKi_RML2 cells were infected with EBOV-Mayinga or MARV-Ozolin at MOI 1.5 and treated with 25 nM BAY-985 for 24 h prior to the addition of 100 ng/mL Chiroptera IFN-β for 30 min. After IFNβ treatment, EBOV-infected cells had elevated cytoplasmic pSTAT1 and significantly reduced nuclear pSTAT1 (90% inhibition) compared to mock infected cells (Fig. 7B, C). Although the levels of cytoplasmic pSTAT1 was decreased significantly (25% inhibition), the amount of nuclear pSTAT1 did not differ between MARV- and mock-infected cells after IFN-β addition (Fig. 7B, C).

To evaluate the importance of MARV VP40 impaired signaling to viral replication, IFN-I signaling was inhibited. The effectiveness of JAK1 inhibitor itacitinib was confirmed first in AjKi_RML2 cells stimulated with 100 ng/mL IFN-β for 30 min (Supplementary Fig. 7B) or transfected with poly(I:C) for 16 h (Supplementary Fig. 7C). AjKi_RML2 cells were infected with EBOV-Mayinga or MARV-Ozolin at a MOI 0.1 for 1 h before addition of itacitinib (0.25–25 nM) in fresh medium. Medium with inhibitor was replaced fully at 24 and 48 HPI, and viral titer in the supernatant at 72 HPI was measured. The MARV titer increased in a dose-dependent manner, but itacitinib treatment did not impact EBOV replication relative to infected cells treated with the vehicle control, DMSO (Fig. 7D). In correlation with the increased MARV titer, elevated VP40 protein expression was itacitinib dose-dependent (Fig. 7E, F). EBOV VP40 protein levels remained relatively constant and were independent of the itacitinib concentration (Fig. 7E, F).

The IFN-I sensitivity of EBOV and MARV in JFB and ERB cells was tested to assess whether bat ISGs differ in their capacity to block filovirus replication. AjKi_RML2 and RoNi cells were treated with IFN-β (0–100 ng/mL) for 24 h prior to infection with MOI 0.1 of EBOV-Mayinga or MARV-Ozolin. At 48 HPI, the IFN-I sensitivity of EBOV and MARV was similar on JFB cells (Fig. 7G), but EBOV is more sensitive to IFN-I than MARV in ERB cells (Supplementary Fig. 7D). IFN-β treatment resulted in the dose-dependent increase of ISG mRNAs in JFB (Supplementary Fig. 7E) and ERB cells (Supplementary Fig. 7F) as expected.

## Discussion

Despite the mounting evidence that bats naturally host orthoebolaviruses, the lack of a full-length genome sequence or isolate for all species that cause disease in humans limits our understanding of which bat species may serve as reservoir hosts. In support of the hypothesis

that bat species naturally host EBOV, we demonstrated that JFB are robustly susceptible to EBOV. Like the recently reported Angolan free-tailed bat model[42], we observed EBOV replication and oral shedding while MARV infection is transient and quenched rapidly. In contrast, ERBs are susceptible to MARV but not orthoebolaviruses[2]. These results suggest that bat species have inherent differences in their susceptibility to different filovirus species. Future studies evaluating JFB infections with other filovirus species will enhance understanding of the within-host factors that dictate virus-host complementarity.

The correlation of the in vitro data with the observed JFB infection outcomes with EBOV and MARV supports that phylogenomics may be useful in predicting which bat species may support a given filovirus species. In support of the decreased entry of MARV pseudoparticles and VSV- MARV-GFP compared to the EBOV GP particles, JFB NPC1, the receptor for EBOV and MARV[43–45], has amino acid differences in the loops that interact with filovirus GP. Future studies using recombinant, mutant JFB NPC1 or filovirus GPs can be used to evaluate the contribution of specific residues in entry. Compared to EBOV, MARV inefficiently blocked the IFN-I pathway in JFB cells. Although the phenotype should be investigated in dendritic cells and macrophages at timepoints following early infection, the impaired immune antagonism of MARV likely contributes to the poor replication and dissemination in vivo. The timing of viral replication and innate immune activation likely contributes to different infection outcomes. Evaluation of transcription with both transcriptomics and RT-qPCR, we observed that the JFBs generate a robust innate immune response toward both EBOV and MARV. Unlike other species, the proinflammatory response is well-controlled which may contribute toward limited pathology. Since MARV replication significantly lags compared to EBOV (2 $\log_{10}$ TCID$_{50}$/mL) in vitro before the IFN-I response is induced, budding, replication, and/or transcription may be attenuated. Evaluation of replication and transcription in JFB cells using the minigenome system is not technically feasible currently, but we are actively working to improve transfection protocols to allow direct evaluation of polymerase functional differences between EBOV and MARV. Adaptation of MARV to JFBs could reveal key mutations needed to enable susceptibility. Importantly, ERBs, despite supporting MARV but not orthoebolaviruses in vivo, do not have differential replication kinetics in vitro. However, we observed that MARV is insensitive to IFN-I in ERB cells relative to EBOV. This suggests that MARV-ERB co-evolution may have enabled MARV to evade the antagonistic functions of ISGs. Different stages in the filovirus life cycle are likely critical in determining virus-host complementarity among bat species. Studying the kinetics of filoviruses in cell lines, preferably representative of different bat families, can contribute knowledge of key innate host factors that influence host susceptibility.

Like other bat-filovirus infection models[1,2,42], EBOV-infected JFBs did not experience morbidity or mortality. In the acute phase, we observed minor-to-mild signs of disease including lethargy, hypothermia, and weight loss (≤5%) in some of the EBOV-infected bats.

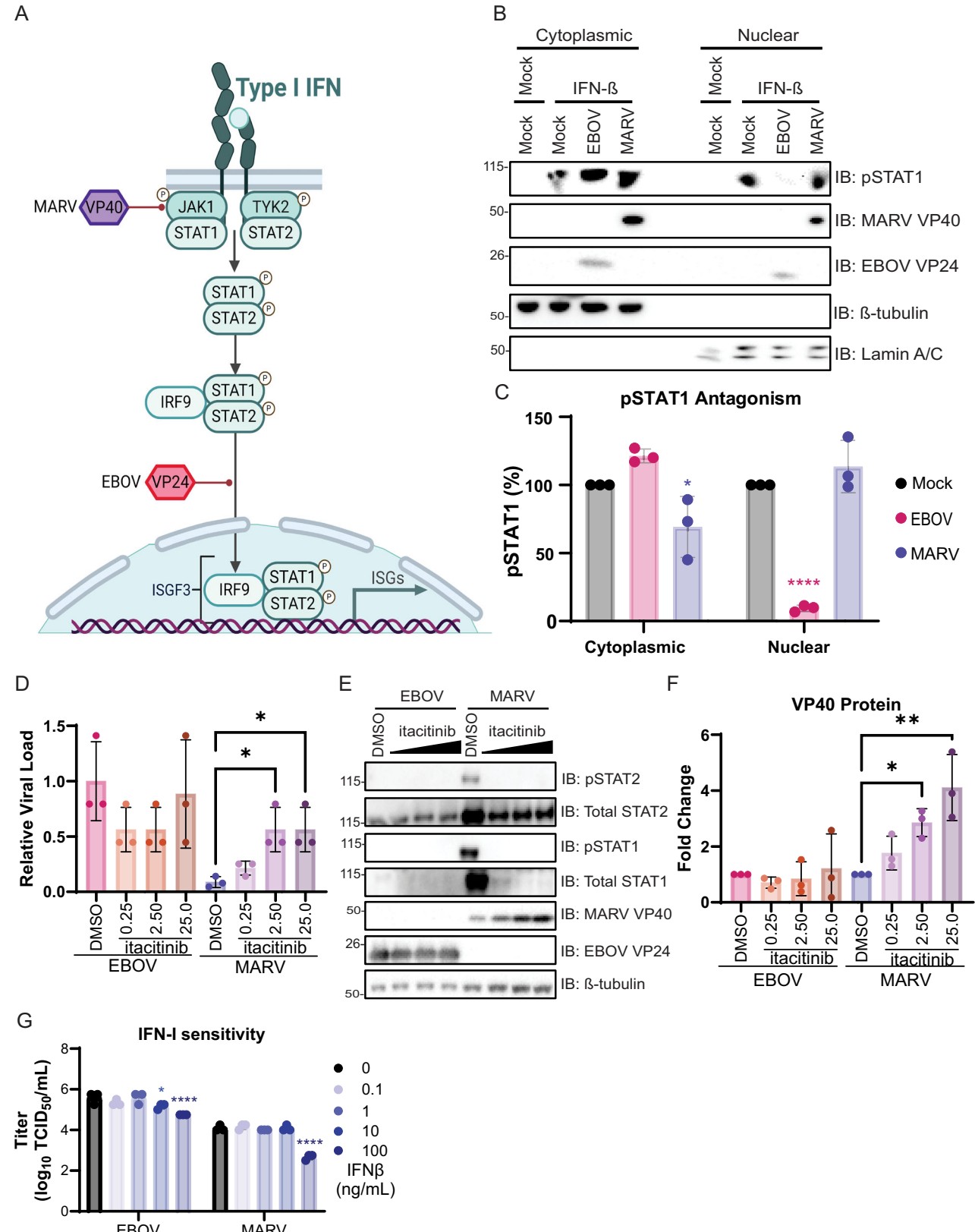

Although all EBOV-infected bats followed through 28 DPI had 2–3 °C decrease in body temperature peaking at 10 DPI that steadily recovered to within 0.5 °C of baseline, we cannot exclude that transponder migration could have contributed to the temperature fluctuations. The infection did not result in any gross lesions and only minor histopathologic changes at the time points evaluated. Unlike infections in dead-end hosts, which can present with hemorrhagic disease[31,32], EBOV infection did not result in lymphopenia or neutrophilia in JFBs. EBOV infection increased the number of circulating monocytes suggesting induction of an early inflammatory response. Akin to other bat-filovirus models[5,46], JFBs mounted a potent IFN-I response while changes in expression of pro-inflammatory cytokine genes was mild.

**Fig. 7 | Inability of MARV to antagonize IFN-I signaling partially contributes to attenuated replication. A** Schematic mechanisms of EBOV VP24 and MARV VP40 antagonism of the type-I interferon signaling pathway. Image created in BioRender [https://BioRender.com/b428467]. **B** Immunoblot of cytoplasmic and nuclear fractions of Jamaican fruit bat kidney cells, AjKi_RML2, infected with MOI 1.5 of EBOV-Mayinga or MARV-Ozolin for 24 h prior to the addition of recombinant Chiroptera IFN-β. The presented panel is representative of three independent immunoblots. **C** Quantification of phosphorylated STAT1 in the cytoplasmic and nuclear fractions. Three independent experiments were performed. Percentage of pSTAT1 in each of the fractions for EBOV and MARV-infected cells was compared mock using a two-way ANOVA with Dunnett's multiple test comparison. Data plotted as mean ± S.D. $P$ values for comparisons to mock are indicated. **** < 0.001, * < 0.05. **D** AjKi_RML2 cells were infected with MOI 0.1 EBOV-Mayinga or MARV-Ozolin for with vehicle control DMSO or itacitinb immediately after infection. The experiment was performed in triplicate. The effect of itacitinb on EBOV and MARV replication was determined using a one-way ANOVA with Dunnett's multiple test comparison. Data plotted as mean ± S.D. $P$ values for comparisons to mock are indicated. * < 0.05. **E** Immunoblot showing the effect of itacitinib treatment on the activation of the immune response and VP40 expression. **F** Quantification of VP40 for the immunoblot presented in (**D**). The effect of itacitinb on EBOV and MARV VP40 expression was determined using a one-way ANOVA with Dunnett's multiple test comparison. The data is from three independent western blots. Data plotted as mean ± S.D. **G** Infectious titers of EBOV-Mayinga or MARV-Ozolin at 48 h post-infection. AjKi_RML2 cells were treated with recombinant Chiroptera IFN-β in 10-fold serial dilution for 24 h prior to infection with MOI 0.1. Data presented as $\log_{10}$ transformed values. The experiment was performed in triplicate. To test a change in viral titer at each dose compared to mock treated cells, we performed a two-way ANOVA with Dunnett's multiple test comparison. Data plotted as mean ± S.D. $P$ values for comparisons to mock are indicated. **** <0.001, * <0.05.

Although the underlying mechanisms in the JFB-EBOV system still need to be evaluated, the lack of a robust inflammatory cytokine response could be tied to differences in inflammasome function[47–50] or STING activation[51] observed in other bat species. Further evaluation of ISGs that are involved in the negative regulation of IFN-I and innate antiviral responses is needed to understand whether their expression contributes to the control of pro-inflammatory responses. Guito and co-authors demonstrated in the ERB-MARV model that dexamethasone-induced immune suppression bolstered MARV replication and shedding and was associated with gross lesions[52]. Although the EBOV-infected JFBs all survived infection, we hypothesize that inoculation with a higher dose or immune suppressing the animal prior to infection could result in a lethal outcome for some individuals.

The JFB-EBOV model is amenable to investigating the factors that influence viral shedding. Infectious EBOV in oral swabs peaked at 6 DPI. These data resemble those reported for the MARV-ERB model in both magnitude and timing[1]. The variation in the amount of virus shed was substantial, suggesting within-host and experimental factors could influence shedding[53,54]. In future studies, we plan to inoculate bats via a single route to understand how exposure route may influence viral dissemination and shedding. Manipulating the bats' diet to mimic nutritional stress or altering climatic conditions could be used to elucidate the impact of environmental stressors on shedding kinetics as well as ISG and cytokine expression. Pregnancy infection models could also be important to address the influence of reproductive status on infection outcome and vertical transmission. Evaluation of the potential for horizontal transmission is also of interest considering that JFBs infected with H18N11 bat influenza virus (IAV) robustly transmitted infectious virus to naïve sentinels[55]. In the Angolan free-tail bat-EBOV model, vertical transmission was documented[42]. Although we detected EBOV RNA in ovary and testis samples collected at 3 and 7 DPI, whether vertical transmission can be achieved in the EBOV-JFB model needs to be investigated.

Establishment of this JFB-EBOV model allows us to interrogate a plethora of open questions in the field. A primary area of concern is the contribution of the host's immune system in controlling infection. Early in infection, dendritic cells and/or macrophages provide an environment for filoviruses to replicate before dissemination[31,32,56]. Future studies will investigate the early innate responses at the inoculated site in response to EBOV and MARV infection to evaluate host factors that may predict infection outcome. Compared to mice that tolerate infection with mouse-adapted EBOV[57], infected JFBs have a strong antiviral transcriptional profile in the liver 3 DPI. Beyond the innate response, the adaptive immune system is critical for controlling infection. In dead-end hosts, filoviruses can infect T lymphocytes and cause cell death through abortive infection[58]. Whether or not filoviruses infect bat T cells remains unknown and could offer insight into differences between lethal and non-lethal infection. Comparing the infections between EBOV and MARV can also provide information of the establishment of an adaptive, humoral response in bats as we observed a MARV GP-specific antibody response in only one of the four infected bats. Although we observed expansion of B cell clones following EBOV infection, the sample size was too small to robustly examine clones within the B cell repertoire that may be EBOV specific. In contrast to JFBs infected with H18N11 IAV which produce robust virus-specific, neutralizing antibodies[59], the neutralizing antibody responses generated in response to filovirus infection were weak. Further development of reagents to study JFBs' adaptive immune response is important to understand better factors that influence the quality of antibody and T cell responses.

Another prevalent question in the field of bat-associated viruses is whether virus persists and recrudesces when a chronically infected bat encounters different stressors. At 28 DPI, we detected EBOV RNA in the spleen at a similar magnitude to 3 and 7 DPI. Due to insufficient sample, we were unable to titrate spleen homogenates collected at 28 DPI and cannot determine whether the virus present was infectious or not. Future studies can address the potential for persistence of infectious virus. In a transmission study using the ERB-MARV model, Schuh and co-authors observed transmission after the acute phase of infection which suggests viral persistence may be a feature of filoviruses[3]. Further, human survivors of Ebola virus disease can have persistent viral RNA and shed infectious virus[60–63].

Overall, JFBs are a valuable model for interrogating filovirus biology. Differential susceptibility to EBOV and MARV offers an opportunity to explore the intricacies of virus-host relationships. Support of EBOV replication with disseminated infection and shedding of infectious virus enables the evaluation of both biotic and abiotic factors that regulate shedding and transmission.

# Methods
## Ethics statement
All animal experiments were conducted in an AAALAC International-accredited facility and were approved by the Rocky Mountain Laboratories (RML) Animal Care and Use Committee, protocol number 2022-029-E, and adhered to the guidelines put forth in the Guide for the Care and Use of Laboratory Animals 8th edition, the Animal Welfare Act, United States Department of Agriculture and the United States Public Health Service Policy on the Humane Care and Use of Laboratory Animals. Work with infectious EBOV and MARV under Biosafety level 4 (BSL4) conditions was approved by the Institutional Biosafety Committee (IBC) and conducted in RML's BSL4 facility. For the removal of specimen from BSL4, virus inactivation of all samples was performed according to IBC-approved standard operating procedures (SOPs).

## Viruses
EBOV-Mayinga (NC_002549; passaged twice since receiving the CDC stock from UTMB) and MARV-Ozolin (AY358025.2; passaged once

since receiving from UTMB) were used for infection studies and in vitro experiments. EBOV-Kikwit (KT582109.1; passaged once since receiving from USAMRIID) and EBOV-Makona C05 (KP096420.1; passaged once since receiving from CSCHAH) as well as MARV-Angola (KY047763.1; passage two) and MARV-Musoke (DQ217792.2, passaged once since receiving from USAMRIID) were also used for in vitro studies. All viruses were sequence verified and confirmed mycoplasma negative. Replication-competent recombinant vesicular stomatitis viruses encoding a green fluorescent protein reporter gene (VSV-GFP) were used for in vitro studies. The generation of VSVwt-GFP and VSV-EBOVmayinga-GFP have been published[64]. The VSV-MARVozolin-GFP was generated following two cloning steps. First, the MARV-Ozolin glycoprotein (GP) gene (AY358025.2) was cloned into the VSV backbone inserted into the VSV backbone (pATX-VSVdeltaG-XN2) between the VSV-M and VSV-L genes replacing VSV-G using Mlu I and Avr II[64]. Next, the GFP gene was added as an additional ORF between the MARV-Ozolin GP and the VSV-L genes. The virus was recovered from by co-transfection of co-cultured Vero and 293T cells with helper plasmids encoding T7 polymerase, VSV L, VSV nucleoprotein, and VSV phosphoprotein for 72 h before transferring the supernatant onto fresh Vero cells[65]. The supernatant was clarified with centrifugation and aliquoted. Protein expression in the supernatant was verified by western blot analysis using antibodies against MARV GP (clone 127-8, 1:1000[66];) and VSV-M (23H12, 1:1000; Kerafast, Inc.).

Virus was propagated in Vero E6 cells (ATCC CRL-1586) in DMEM supplemented with 2% fetal bovine serum (FBS), 1 mM L-glutamine, 50 U/mL penicillin, and 50 µg/mL streptomycin (DMEM2). Vero E6 cells were maintained in DMEM supplemented with 10% FBS, 1 mM L-glutamine, 50 U/mL penicillin, and 50 µg/mL streptomycin (DMEM10). No *Mycoplasma* nor contaminants were detected in any of the virus stocks.

## Cells

JFB, or *Artibeus jamaicensis* (Aj), cells including immortalized and primary kidney (AJi and AjKi_RML2, respectively), lung (AjLu_RML3), and uropatagium derived fibroblasts (AjUFi_RML6) and ERB, or *Rousettues aegyptiacus* (Ra), cells including primary kidney (RASKM), and primary lung (RaLu1.4)[67] were maintained in DMEM/F-12 supplemented with 1X MEM non-essential amino acids (NEAA) (Gibco), 10% FBS, 50 U/mL penicillin, and 50 µg/mL streptomycin (10DMEM/F-12). Vero E6 (CRL1586), immortalized ERB kidney (RoNi), and immortalized human hepatoma (Huh-7) cells were maintained in DMEM10. All cells were maintained at 37 °C, 5% $CO_2$ and were *Mycoplasma* negative.

## Bat inoculation

Jamaican fruit bats from a closed colony housed at Colorado State University were used in this study. Upon arrival at RML, bats were co-housed in same sex groups of up to six animals per stainless steel cage. Bats were fed fresh fruit twice daily and provided water. The study comprised 28 animals total, 14 males and 14 females. Bats were allowed to acclimate to the facility for 5 days before collecting samples. Bats were intranasally, orally, and subcutaneously inoculated with a total of $3.0 \times 10^4$ $TCID_{50}$ of EBOV-Mayinga (six females (EBOV01-06) and six males (EBOV07-12)) or MARV-Ozolin (six females (MARV01-06) and six males (MARV07-12)). Bats that arrived with the same shipment of bats were used as healthy controls for histologic (2 males, 2 females), B cell receptor (BCR) sequencing (5 males, 5 females), and gene expression (2 males, 2 females) analyses. All inoculations and subsequent manipulations were performed under isoflurane (1–5%) anesthesia. Four EBOV- and four MARV-inoculated bats were euthanized at 3-, 7-, and 28-days post-inoculation (DPI) to assess viral replication and host gene expression in tissue samples. Oropharyngeal and rectal swabs were collected to monitor viral shedding prior to infection, every 2 days after infection through 14-DPI, 21-DPI, and at necropsies. Blood was collected at baseline, on necropsy days, and on 14 and 21 DPI. At necropsy, whole blood

was used for hematology, and serum was used for clinical chemistry. Hematology analysis was completed on a ProCyte DX (IDEXX Laboratories, Westbrook, Maine). Serum chemistries were completed on a VetScan VS2 Chemistry Analyzer (Abaxis, Union City, California). Bats' body temperatures were obtained via implanted transponder (BMDS IPTT-300) and weights were taken on days -8/7, 2, 4, 6, 8, 10, 12, 14, 17, 21, 24, and 28. Swabs were collected in 1 mL DMEM2. Bats were observed daily for clinical signs of disease. Necropsies and tissue sampling were performed according to IBC-approved SOPs.

## In vitro infections

To assess filovirus replication and host gene expression kinetics, AjKi_RML2 and RoNi cells were plated at 250,000 cells/mL in 10DMEM/F-12 or DMEM10, respectively, in 24-well plates and incubated overnight at 37 °C, 5% $CO_2$. The next day, medium was removed, and cells were inoculated with either EBOV-Mayinga or MARV-Ozolin at multiplicity of infection (MOI) 0.1 for 1 h at 37 °C, 5% $CO_2$ with rocking every 15 min. Cells were then washed three times with 1X Dulbecco's phosphate buffered saline, no calcium, no magnesium (DPBS), and 0.5 mL of fresh medium, DMEM/F-12 with 2% FBS, 1X NEAA, and 50 U/mL penicillin, and 50 µg/mL streptomycin (2DMEM/F-12) for AjKi_RML2 and DMEM2 for RoNi, was added. A sample of the medium was collected at the 1-h time point, and the volume was replaced with fresh medium. Every 24 h for 5 days, 500 µL of supernatant was collected for titration and the cells were lysed in 600 µL of RLT (QIAGEN) for RNA extraction or 4% sodium dodecyl sulfate (SDS) buffer with 10% β-mercaptoethanol for western blotting. Protein samples were boiled at 100 °C for 10 min. Both supernatant and RLT samples were stored at −80 °C until processing. The same protocol was used with EBOV-Kikwit and EBOV-Makona C05 and MARV-Angola and MARV-Musoke in additional JFB and ERB cell lines at 72 and 120 HPI.

To evaluate antagonism of IFN-I signaling, AjKi_RML2 and Huh-7 cells were plated at 250,000 cells/mL in 6-well plates in 2.0 mL of the appropriate medium and incubated overnight at 37 °C, 5% $CO_2$. The next day, medium was removed, and cells were inoculated with MOI 1.5 of virus (EBOV-Mayinga or MARV-Angola) for 1 h at 37 °C, 5% $CO_2$ with rocking every 15 min. After incubation, the inoculum was removed, and 2.0 mL of fresh medium was added and cells were returned to the incubator for 24 h. Cells were stimulated with 100 ng/mL of Chiroptera IFN-β for 30 min or mock treated with equal volume of medium. Cells were trypsinized for 10 min with TrypLE Express (Gibco) and spun down after addition of an equal volume of complete medium. After washing the cell pellet once with 1X DPBS, the cytoplasm and nucleus were fractionated using NE-PER™ Nuclear and Cytoplasmic Extraction Reagents (Thermo Scientific, Cat # 78833) according to the manufacturer's instruction. The nuclear and cytoplasmic fractions were inactivated according to SOP and western blot analysis was performed as described below.

## IFN-β expression and purification

A bat *Ifnb1* consensus sequence was designed using the *Ifnb1* gene sequence from all bat genomes available in the NCBI database as of November 2022. To prevent bias from the number of species represented in different bat families, we first generated a consensus of each bat family and generated the Chiroptera consensus from the bat family consensus sequences. An IL-2 secretion signal, His-tag, and TEV protease signal were added upstream of the consensus *Ifnb1* (Chiroptera *Ifnb1*) open reading frame and the synthesized sequence was inserted into the pcDNA3.4 vector (GenScript).

tsA201 cells derived from HEK293 (Millipore Sigma, cat# 96121229-1VL), from which *Mycoplasma* was eradicated at ECACC, and the identity of tsA201 and 293 has been confirmed by STR (short tandem repeat) profiling. Cells were adapted to grow in suspension by using Gibco® FreeStyle™ 293 Expression Medium plus 2 mM glutamine in a shaker flask. The low passage (P2 or P3) of the suspension culture

was collected, and cell stock was made with the addition of 7.5% DMSO and then stored in liquid nitrogen. For each expression, one frozen vial with $1 \times 10^7$ tsA201 cells was inoculated with 40 mL of the above-mentioned medium in a 250 mL flat-bottom cell culture shaker flask. Cell density was measured after cells grew for 3 days in an Infors Multitron incubator at 130 rpm with 8% $CO_2$ and 75% humidity, using NucleoCounter® NC-100™. Cells were then passed into a large flask (2.8 L or 5 L Optimum Growth Flask) with the addition of the above-mentioned medium to reach a final cell density of $1.3 \times 10^5$/mL. The total volume of culture was no more than 1.0 L for a 2.8 L flask and no more than 2.1 L for a 5 L flask. Transfection was started when, cells reached a density of $2 \times 10^6$/mL, typically after 3 days. For one liter of culture, 1 mg of pcDNA3.4-Chiroptera-*Ifnb1* plasmid (filtered with a 0.22 μm filter) was added to 50 mL of prewarmed Hybridoma-SFM (ThermoFisher, 12045076). Four milliliters of PEI MAX® (Transfection Grade Linear Polyethylenimine Hydrochloride, MW 40,000, Polysciences, 24765-1) at 1 mg/mL was added, then gently mixed to homogenize. After incubation at room temperature for 12–15 min, the mixture was added to the culture, and then the flask was returned to the incubator. After 5 days, the culture was centrifuged at $1000 \times g$ for 15 min using a Sorvall RC3B plus centrifuge. The supernatant was stored at −80 °C for future purification.

The supernatant was filtered through a 0.22 μm filter, then mixed with 5 mL of prewashed HisPur™ Ni-NTA Resin (ThermoFisher Scientific) and rocked overnight at 4 °C. The following day, the supernatant was drained, and the resin was packed into a 5 mL column. AKTA FPLC with Unicorn software (GE Healthcare) was used to wash the column with sodium phosphate buffer, pH 7.0, and 200 mM NaCl at 2 mL/min. The protein was eluted with a linear gradient up to 400 mM imidazole over 30 CV. The IFN-β-containing fractions were pooled (~28 mL) and injected into a 30 mL Slide-A-Lyze cassette with 10 K MWCO (ThermoFisher Scientific). The cassette was then dialyzed against 1.8 L of 50 mM sodium phosphate buffer, pH 7, three times. After dialysis, the protein concentration was measured using a NanoDrop Microvolume Spectrophotometer (ThermoFisher Scientific). IFN-β was then cleaved by TEV protease to remove the N-terminal His tag. The enzymatic cleavage was completed with 1 mg of TEV per 5 mg of IFN-β and incubated at 4 °C overnight with rocking. The digested mixture was then loaded onto a pre-equilibrated 5 mL Ni-NTA Superflow column (Qiagen) to remove the cleaved His-tag, remaining uncleaved protein, as well as additional impurities, with a flow rate of 1 mL/min. The flow-through fractions were loaded onto a pre-equilibrated HiTrap Capto Q anion exchange column (Cytiva) to separate IFN-β aggregates from monomer with a flow rate of 2 mL/min. The running buffer was 50 mM sodium phosphate, pH 7, and the gradient was 5 mM/min for NaCl from 0 to 500 mM. IFN-β monomer-containing fractions around 200 mM NaCl were immediately collected and analyzed with SDS-PAGE.

### IFN-I pathway inhibition

AjKi_RML2 cells were treated with 5-fold serial dilution of IFN-I induction (BAY-985), signaling (deucravacitinib, fludarabine, or itacitinib) inhibitors or dimethyl sulfoxide (DMSO) vehicle control for 24 h prior to stimulation with 500 ng/mL high molecular weight poly (I:C) (Invitrogen) for 18 h (BAY-985) or 100 ng/mL recombinant consensus bat IFN-β (IFN-I signaling inhibitors) for 30 min. Protein was collected after stimulation and the effect of blocking IFN-I signaling on filovirus replication was assessed after infection of cells with MOI 0.1 of EBOV-Mayinga or MARV-Ozolin, as described above, then added itacitinib in 10-fold serial dilution (0.25–25 nM) or equal volume of DMSO. The medium was replaced with fresh inhibitor every 24 h before collection of supernatants for titration at 72 h post-infection.

### IFN-β sensitivity

To assess the sensitivity of EBOV and MARV to IFN-β, AjKi_RML2 or RoNi cells were treated with 10-fold serial dilutions of Chiroptera IFN-β

(0.1–100 ng/mL) for 24 h prior to infection with MOI 0.1 of EBOV-Mayinga or MARV-Ozolin. Uninfected cells were lysed in RLT 24 h after IFN-β stimulation to measure the induction of interferon stimulated genes at the time of infection. Supernatants were collected 48 h after infection and titrated on Vero E6 cells as described above.

### Western blot analysis

Inactivated cell lysates collected in SDS buffer were run on 4–12% Bis-Tris NuPAGE gels (Invitrogen) at 150 V for 1 h then transferred onto methanol-activated PVDF membrane (BioRad). Following a 1 h block in 5% powdered milk, the membranes were washed in buffer (1X tris-HCL with 0.1% tween 20) three times. Membranes were probed with primary antibody overnight rocking at 4 °C. The next day, blots were washed three times, incubated with an HRP-conjugated secondary antibody for 1 h rocking at room temperature, and washed three times before development. Blots were incubated with a 1:1 ratio of peroxidase and enhancer reagents Clarity Western ECL (BioRad) or SuperSignal West Femto (Thermo Scientific) and developed on an iBright imaging system (Thermo Fisher Scientific). To calculate relative expression, area under the curve was determined for each band using Fiji[68]. Primary antibodies used: 1:1000 pSTAT1 – Y701 (Cell Signaling Technology, 9167S), 1:1000 pSTAT2 – Y690 (Cell Signaling Technology, 88410S), 1:1000 total STAT1 (Cell Signaling Technology, 14994S), 1:1000 total STAT2 (Cell Signaling Technology, 72604S), 1:1000 RIG-I (Kerafast, 1C3), 1:1000 Lamin A/C (Cell Signaling Technology, 4777S), 1:1000 β-tubulin (Sigma Aldrich, T8328), 1:1000 EBOV VP24 (Sino Biological 40454-T46), 1:1000 EBOV VP40 (GeneTex, GTX134034), 1:1000 MARV VP40 (The Native Antigen Company, MAV12450-100). Secondary antibodies used: 1:10,000 Donkey-anti-rabbit (GE Healthcare, NA934) and 1:10,000 Sheep-anti-mouse (GE Healthcare, NA931). Area under the curve (AUC) for western blot bands was measured with Fiji[68]. Normalized expression was determined with phosphorylated protein with the following formulas: (AUC pSTAT1/AUC β-tubulin or Lamin A/C) or (AUC VP40/ AUC β-tubulin). To calculate relative expression, the normalized expression of the sample is divided by the normalized expression of the mock/non-infected value. The uncropped western blot images are provided in the source file.

### Pseudoparticle assay

Pseudoparticles were generated through transfection of 293T cells with plasmid encoding T7 polymerase, a VSV plasmid with GFP replacing G (VSVΔG), VSV nucleocapsid, VSV phosphoprotein, VSV polymerase, and plasmids encoding EBOV-Mayinga GP or Marburg-Musoke GP[69]. Infectious units (IUs) of these pseudotyped VSVs were determined on Vero E6, AjKi_RML1, AjKi_RML2, and RoNi cells by titrating the replication incompetent viruses and measuring the GFP positive cells 16–24 h after infection[69]. The normalized entry ratio was calculated for each cell line using the following formula: ((IU EBOV/IU MARV) for cell line of interest)/((IU EBOV/IU MARV) for Vero E6).

### Recombinant VSV infections

Stocks of replicating recombinant VSV (rVSV)-GFP encoding GPs from wild-type VSV, EBOV-Mayinga, or MARV-Ozolin were titrated on either Vero E6 cells or AjKi_RML2 to determine differential TCID$_{50}$. To evaluate replication kinetics, Vero E6 or AjKi_RML2 cells were infected at MOI 0.005 with VSV-MARVozolin-GFP and -EBOVmayinga-GFP for 1 h at 37 °C, 5% $CO_2$ with constant rocking. Cells were washed three times with 1 mL of DPBS before the addition of fresh medium. The supernatants were titrated on Vero E6 cells.

### RNA extractions and RT-qPCR

Swabs and blood were collected as described above; 140 μL was used for RNA extraction using the QIAamp Viral RNA Kit (Qiagen) according to the manufacturer's instructions with an elution volume of 60 μL. For tissues and cells, RNA was isolated using the RNeasy Mini kit (Qiagen)

according to the manufacturer's instructions and eluted in 50 µL. Viral RNA was detected by RT-qPCR (Supplementary Table 5). RNA was tested with TaqMan™ Fast Virus One-Step Master Mix (Applied Biosystems) using QuantStudio 6 or 3 Flex Real-Time PCR System (Applied Biosystems). EBOV or MARV standards with known copy numbers were used to construct a standard curve and calculate copy numbers/mL or copy numbers/g.

RLT lysates from cell monolayers were transferred to 70% ethanol prior to extraction of RNA using the RNeasy extraction kit (Qiagen). Cellular RNA was used to measure viral RNA and host genes using RT-qPCR (Supplementary Table 5). Following extraction, 17 µL of RNA was treated with TURBO DNase (Invitrogen) according to the manufacturer's instructions. After DNase treatment, the RNA was diluted 1:5 in molecular grade water (Invitrogen). DNase-treated RNA (5 µL) was used for each host gene assessed. Fold change was calculated for host genes by dividing the sample relative gene expression by the average relative gene expression of the healthy controls or mock-treated cells ($2^{-\Delta\Delta Ct}$).

## Virus titration
Infectious virus in tissue, swab, and supernatant samples was evaluated in Vero E6 cells. Tissue samples were weighed, then homogenized in 1 mL of DMEM2. Vero E6 cells were inoculated with ten-fold serial dilutions of homogenate, swabs, blood, or supernatant incubated for 1 h at 37 °C. For tissue homogenates and blood, the first two dilutions of each sample replicate were washed twice with DMEM2. For swab and supernatant samples, cells were inoculated with ten-fold serial dilutions and no wash was performed. On days 10–14, cells were scored for cytopathic effect (CPE). Titers in $TCID_{50}$/mL were calculated by the Spearman-Karber method.

## Serology
Maxisorp plates (Nunc) were coated with 50 ng of EBOV-Mayinga[70] or MARV-Angola (IBT Bioservices, cat no. 0506-015) GP per well. Plates were incubated overnight at 4 °C. Plates were blocked with casein in phosphate buffered saline (PBS) (ThermoFisher) for 1 h at room temperature. Sera were diluted 1:100 in blocking buffer and samples (triplicate) were incubated for 1 h at room temperature. Secondary horseradish peroxidase (HRP)-conjugated recombinant protein-A/G (Invitrogen, lot number WH 324034) diluted 1:10,000 was used for detection and visualized with KPL TMB two-component peroxidase substrate kit (SeraCare, 5120-0047). The reaction was stopped with KPL stop solution (SeraCare) and plates were read at 450 nm. Plates were washed three times with PBS-T (0.1% Tween) between each step. Arbitrary enzyme-linked immunosorbent assay (ELISA) units were calculated based on the standard curve. All samples diluted at 1:100 with an optical density value of <0.250 were given a value of 1.

## Virus neutralization
Heat-inactivated, irradiated sera were two-fold serially diluted in DMEM2, and 100 $TCID_{50}$ of EBOV-Mayinga or MARV-Ozolin was added. After 1 h of incubation at 37 °C and 5% $CO_2$, the virus: serum mixture was added to Vero E6 cells. CPE was scored after 11–14 days incubated at 37 °C and 5% $CO_2$. Viral neutralization was determined based on the dilution factor non-CPE wells were observed.

## Flow cytometry
Bat cells were isolated from spleens by generating a single cell suspension using a 100-µm filter and red blood cells were removed using RBC lysis buffer (Invitrogen). Splenocytes were then cryopreserved in freezing medium (10% DMSO and 90% FBS) and stored at −80 °C. The cryopreserved bat splenocytes were rapid thawed in RPMI-1640/10% FBS and washed twice. To test for presence of EBOV GP-specific B cells, Alexa Fluor 568 was conjugated to recombinant EBOV GP receptor binding domain (Sino Biological, 40304-V08H) using an Invitrogen

Alexa Fluor protein labeling kit (Invitrogen, A10238). They were first stained with ViaKrome 808 Fixable Viability Dye (Beckman Coulter) for 30 min at room temperature (RT) and then surface stained with EBOV-GP AF568 for 30 min at RT. Samples were then fixed and permeabilized using a Cytofix/Cytoperm kit (BD Biosciences). The antibodies used for intracellular staining were: 1:100 Allophycocyanin-anti-CD79a (HM57, BD Biosciences 752115) and 1:100 Fluorescein isothiocyanate-anti-CD3ε (CD3-12, BioRad MCA1477F). Samples were analyzed on a 6-laser Cytoflex LX (Beckman Coulter) and flow cytometry data were analyzed using FlowJo v.10.9 (BD Life Sciences). Live lymphocytes were gated using a FSC-A vs Live-Dead gate, followed by a FSC-A and FSC-H gate to exclude doublets, and then by a standard FSC-A vs SSC-A lymphocyte gate (Supplementary Fig. 8).

## B cell receptor sequencing
Splenic mRNA was harvested from healthy control JFBs and from EBOV infected bats at 3-, 7-, and 28 DPI. The mRNA was inactivated and stored in Trizol according to institutional SOP. After addition of chloroform to Trizol, RNA was extracted with Phasemaker tubes (Invitrogen). For BCR library preparation, we used JFB IgM and IgG primers (Supplementary Table 5) for bat tissue[54]. We prepared BCR libraries with the NEBNext® Ultra II DNA Library Prep Kit with Sample Purification Beads (cat# E7103S) and NEBNext® Multiplex Oligos for Illumina® (Index Primers Set 1, cat# E7335S). Samples were cleaned using NebNext Ultra II beads and enriched for libraries with a size range of 500–700 base pairs amplicons, according to manufacturers instructions[71]. All libraries were sequenced using the Illumina MiSeq 2 × 300 chemistry platform.

Sequencing reads were cleaned using Immcantation version 4.3.0. Sequences were filtered out if they had an average FASTQ score below 30. Sequences with shared unique molecular identifiers were used to generate consensus sequences. Consensus sequences with only one contributing read were removed from the dataset. The *Mus musculus* germline was used to classify V, D, and J mRNA sequences into putative germline genes using IgBLAST (version 1.18.0[72]. The mouse germline was used because the V, D, and J germline genes have not been fully annotated for the JFB.

Consensus sequences were binned into clonal lineages with a Hamming distance cutoff of 0.09. The frequency of clones and $V_H$ and $J_H$ gene segments were assessed using Alakazam (version 1.2.1). Clonal frequencies were normalized with an ordered quantile normalization transformation using the R library BestNormalize (version 1.9.1). For each bat, a subsample of 300 clones was randomly selected. These subsamples were then analyzed using a Bayesian hierarchical model implemented in Rstan (version 2.32.5). The model estimated the average normalized clonal frequency for each time point based on the data from the bats euthanized at those respective time points. The model pooled information across individuals to estimate a population-level average clonal frequency for each time point, while accounting for individual variation.

## Histopathology
Necropsies and tissue sampling were performed according to IBC-approved SOPs. Tissues were collected and fixed for a minimum of 7 days in 10% neutral buffered formalin with two formalin changes. Tissues were placed in cassettes and processed with a Sakura VIP-6 Tissue Tek on a 12-hautomated schedule using a graded series of ethanol, xylene, and PureAffin. Prior to staining, embedded tissues were sectioned at 5 µm and dried overnight at 42 °C. Histopathologic analysis was performed by a blinded, board-certified veterinary pathologist.

## Next-generation sequencing of mRNA
RNA from livers of negative control or 3 DPI EBOV- or MARV-infected bats was used for analysis. Two hundred nanograms of RNA was used

as input for poly-A pull-out and next-generation sequencing (NGS) library preparation following the Illumina Stranded mRNA prep workflow (Illumina). The NGS libraries were prepared, amplified for 15 cycles, purified with AMPureXP beads (Beckman Coulter), assessed on a TapeStation 4200 (Agilent Technologies), and quantified using the Kapa Library Quantification Kit (Illumina). Amplified libraries were normalized, pooled at equal 2 nM concentrations, and sequenced as 2 × 75 base reads on the NextSeq instrument using three high output chemistry kits (Illumina). Raw fastq reads were trimmed of Illumina adapter sequences using cutadapt (version 1.12), and then trimmed and filtered for quality using the FASTX-Toolkit (Hannon Lab). Remaining reads were aligned to the Jamaican fruit bat genome assembly version (GCF_021234435.1) using Hisat2 [35]. Reads mapping to genes were counted using htseq-count [36]. Differential expression analysis was performed using the Bioconductor package DESeq2 [37] and data were further analyzed and plotted using ggplot2 (V3.4.0) as part of the tidyverse package (V1.3.2) [38]. Pathway analysis was performed using Ingenuity Pathway Analysis (Qiagen), and gene clustering was performed using Partek Genomics Suite (Partek Inc.).

### Statistical analysis

Significance tests were performed as indicated where appropriate for the data using GraphPad Prism 10.2.0. Unless stated otherwise, statistical significance levels were determined as follows: no symbol = $p > 0.05$; * = $p \leq 0.05$; ** = $p \leq 0.01$; *** = $p \leq 0.001$; **** = $p \leq 0.0001$. The statistical test used is specified where appropriate.

### Reporting summary

Further information on research design is available in the Nature Portfolio Reporting Summary linked to this article.

## Data availability

The data generated in this study have been deposited in Figshare at https://doi.org/10.6084/m9.figshare.27854799, and the data generated in this study are provided in the Source Data file. The transcriptomics datasets have been deposited to NCBI (PRJNA1219753, [https://www.ncbi.nlm.nih.gov/sra/PRJNA1219753]). The Jamaican fruit bat genome (GCF_021234435.1, [https://www.ncbi.nlm.nih.gov/datasets/genome/GCF_021234435.1/]) was used as the reference to analyze the transcriptomics data. Source data are provided with this paper.

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

## Acknowledgements

We would like to thank the animal caretakers for their assistance with animal husbandry, and Kent Barbian for his assistance with sample processing for transcriptomics. This study was supported by the Intramural Research Program of NIAID, NIH.

## Author contributions

S.v.T., J.R.P. designed the studies. S.v.T., J.R.P., R.J.F., S.G., T.B., A.G., J.E.S., A.C., L.M., D.E.C., C.A.F., J.C.R., A.W., C.C., J. Lovaglio, C.S., G.S., J.P.-S., performed the experiments. S.v.T., J.R.P., J.C.R., L.M., D.E.C., C.C., J. Lack, C.M., S.A. analyzed results. Y.H., T.S., L.V.K., R.K.P., and A.M. provided essential reagents and support. S.v.T., V.M. wrote the manuscript. All co-authors reviewed the manuscript and approved this version.

## Competing interests

The authors declare no competing interests.
