## [Transparent Peer Review file · Nature Communications]

Jamaican fruit bats competence for Ebola virus but not Marburg virus is driven by intrinsic differences in viral entry and IFN-I signaling antagonism

Corresponding Author: Dr Vincent Munster

Version 0:

Reviewer comments:

Reviewer #1

(Remarks to the Author)

This is a very well written paper that describes the exploration of the Jamaican fruit bat as a potential bat model for studying Marburg and/or Ebola virus. The experiments were well designed and executed and the results interpreted correctly. The authors show convincingly that Ebola virus, but not Marburg virus, replicates and is shed from JFBs, and through a series of experiments, including in vitro infections of multiple JFB and ERB cell lines with multiple EBOV and MARV strains, that this difference could be attributed, in part, to differences in IFN-1 signaling antagonism and entry. The findings will be of high interest to the scientific community interested in mechanisms of host competence and the development of potential filovirus bat models. Until the EBOV reservoir (presumably a bat) is identified, this JFB model should be highly valuable and offer multiple lines of investigation.

Minor queries to address:

1. What was the rationale for inoculating the bats simultaneously using multiple routes? Presumably this was to ensure infection but could yield confound result interpretation. The JFB colony is robust, so more numbers could be used to separate oral/intranasal vs SQ unless this reflects a limitation of BSL-4 work. Is there a difference?
2. The use of Ozolin for the JFB infections seems odd given the infrequency of use in other animal models compared to Angola, Popp or Musoke. Why?
3. In Figs 3, 4, 6, it's very difficult to discern the different color coding. It may be a function of lower quality PDFs for review purposes, but the final product needs to be clearer.
4. The discussion, though well written and thoughtful, is light on interpretation of specific data such as IFN antagonism, VDJ clonal data and consequences of the specific immune genes or pathways up/down regulated. Additional discussion would be helpful.
5. Also for the discussion, could the authors briefly comment on how this data compares to any similar analyses done with their JFB flu model.
6. In fig 6F, 6J it's odd that the first time point (24 hr) for EBOV is consistently higher than MARV. Is this reflective of more rapid replication or less virus being washed off during the adsorption step. A zero or 1 hr timepoint would help.
7. The authors hypothesize that high dose inoculation could result in a lethal outcome. Why do the authors think that?
8. For the 28 DPI EBOV+ spleen sample, was virus isolation attempted?

Reviewer #2

(Remarks to the Author)

Reviewer #3

(Remarks to the Author)

Dear Authors, The manuscript unravels the differences in the behaviour of EBOV and MARV in Jamaican fruit bats at both the animal and cellular levels, highlighting in particular the specific relationship between certain filoviruses and specific bat species. The data are of great importance to the scientific community and shed light on the interactions between filoviruses and reservoir species. A few points can be raised to reinforce the message of the articles:

Lane 128: Please indicate the history of virus passage.

Lane 158 : Please provide information on the housing conditions of the bats (surface, diet, anaesthesia).

Lane 304 : The reference refers to another reference.

Lane 380 / Suppl Figure 1 / Figure 4L : please add number of events and % in Suppl Figure 1E and add values for all animals as the plot of S1E is currently not very convincing.

Lane 465 : RNA values are very close to infectious titers, which is not expected for mononegaviruses.

Lane 478 : Please indicate the viral RNA level in the sera of the animals.

Lane 488 : Have the adrenal glands been titered for viral RNA and virus, as filoviruses are usually abundant there in susceptible hosts?

Lane 490 : EBOV and MARV antigen staining would be a real plus, as would caspase 3 staining of spleen sections, as this is a hallmark of filovirus infection in susceptible hosts.

Lane 506 : as reservoir species are supposed to control but allow viral replication, a check for regulating/proviral ISGs would be of interest (SOCS, USP18, ...).

Lane 540 : Anti-GP antibodies were evaluated to signify the immunisation of the animal, but antibodies raised against nucleoproteins are usually more abundant and may have provided a better signature.

Lane 543 : typo animal EBO420, is it EBO06?

Lane 528 : There is more than a one log difference in viral RNA quantity between EBOV and MARV, so the interpretation of transcriptomic data should take this into account.

Lane 611 / Figure 6E, please comment on the global increase in total STAT.

Lane 670 : Why were Huh7 stimulated?

Lane 748 : Please comment on the hypothermia observed?

Version 1:

Reviewer comments:

Reviewer #3

(Remarks to the Author)

Dear authors, thank you for acknowledging most of my concerns, it is sad that IHC is not working properly, I hope to see results in the next article.

made.

Reviewer #1 (Remarks to the Author):

This is a very well written paper that describes the exploration of the Jamaican fruit bat as a potential bat model for studying Marburg and/or Ebola virus. The experiments were well designed and executed and the results interpreted correctly. The authors show convincingly that Ebola virus, but not Marburg virus, replicates and is shed from JFBs, and through a series of experiments, including *in vitro* infections of multiple JFB and ERB cell lines with multiple EBOV and MARV strains, that this difference could be attributed, in part, to differences in IFN-1 signaling antagonism and entry. The findings will be of high interest to the scientific community interested in mechanisms of host competence and the development of potential filovirus bat models. Until the EBOV reservoir (presumably a bat) is identified, this JFB model should be highly valuable and offer multiple lines of investigation.

Minor queries to address:

1. What was the rationale for inoculating the bats simultaneously using multiple routes? Presumably this was to ensure infection but could yield confound result interpretation. The JFB colony is robust, so more numbers could be used to separate oral/intranasal vs SQ unless this reflects a limitation of BSL-4 work. Is there a difference?

As the reviewer suggests, the multiple routes used were to ensure infection of the bats for the initial study. Future studies are planned using individual inoculation routes to understand which allows Ebola virus to establish infection. We do not think it is necessary to include those data in the manuscript since the resulting data would not change the overall conclusion of the manuscript that Ebola virus, but not Marburg virus, can establish disseminated infection in Jamaican fruit bats. We have included a comment in the discussion that future studies using single routes will be helpful to interpret how the virus establishes infection (lines 434-436).

2. The use of Ozolin for the JFB infections seems odd given the infrequency of use in other animal models compared to Angola, Popp or Musoke. Why?

Marburg Ozolin was chosen due to being the most closely related to Marburg bat 371 which is used in the Marburg Egyptian rousette bat model, and we wanted our results to be better comparable to that model. Due to the *in vitro* results with multiple Marburg virus strains, we expect similar results *in vivo* regardless of the strain used. We have included a comment addressing the rationale for using MARV strain Ozolin (lines 100-101).

3. In Figs 3, 4, 6, it's very difficult to discern the different color coding. It may be a function of lower quality PDFs for review purposes, but the final product needs to be clearer.

We have improved the color contrast for the different days post-infection in Figures 3, 4, and Supplementary Figure 3, and we made the EBOV and MARV strains different colors to improve clarity in Figures 6 and Supplementary Figure 4.

4. The discussion, though well written and thoughtful, is light on interpretation of specific data

such as IFN antagonism, VDJ clonal data and consequences of the specific immune genes or pathways up/down regulated. Additional discussion would be helpful.

We have added several points to the discussion on IFN antagonism and the bias toward antiviral versus pro-inflammatory responses (lines 389-393 and lines 422-424) and the VDJ clonal data (lines 460-461). Due to the small sample size, we do not want to overinterpret the VDJ clonal data, so we did not deepen the discussion much.

5. Also for the discussion, could the authors briefly comment on how this data compares to any similar analyses done with their JFB flu model.

We have added discussion on how the JFB-flu model compares to this data for both evaluating transmission (lines 440-442) and the quality of virus-specific antibodies (lines 462-465).

6. In fig 6F, 6J its odd that the first time point (24 hr) for EBOV is consistently higher than MARV. Is this reflective of more rapid replication or less virus being washed off during the adsorption step. A zero or 1hr timepoint would help.

Based on the 1 hour time point in 6A, the difference in virus beginning at 24 hr is due to EBOV consistently replicating to higher titer than MARV. The experiment reported in 6A was replicated both for this manuscript and since, and we always observe equivalent titer at 1 hr and several log higher replication for EBOV compared to MARV at 24 hours. Due to the volume samples collected in parallel for 6 different viruses on 8 cell lines in triplicate, we chose to collect the most important time points, 24, 72, and 120 hours, for Figures 6F and 6J and Supplementary Figure 4G and 4H. These experiments were completed in parallel so that we could compare the replication kinetics on Jamaican fruit bat cells (Figures 6F and 6J) to those on human (Supplementary Figure 4G) and Egyptian rousette bat cells (Supplementary Figure 4H). Since we only observe the difference in titer between EBOV and MARV on the Jamaican fruit bat cells, this is attributable to the more efficient replication of EBOV.

7. The authors hypothesize that high dose inoculation could result in a lethal outcome. Why do the authors think that?

Two of the four bats in the 28 DPI group had weight loss, and one had mild hypothermia in addition to the weight loss. It's not impossible that the hypothermia could be due to migration of the transponder, but due to the trajectory of the temperature we think te change in temperature was a physiological response to infection (lines 410-413). Though there was no formal clinical scoring, the female bats presented as lethargic through day 10. We want to avoid over-interpretation of the data at this point since it's hard to definitely describe sickness behaviors, but we think a higher dose could lead to more dramatic hypothermia and/or weight loss which may result in reaching euthanasia criteria.

8. For the 28 DPI EBOV+ spleen sample, was virus isolation attempted?

Unfortunately, as mentioned in the manuscript (lines 138-139 and 468-469), no we were unable to attempt virus isolation due to insufficient sample. The spleen was used for histology, qPCR, and flow cytometry, so we did not have an additional piece for virus isolation. In future studies, we plan to evaluate the presence of infectious virus to understand potential of persistent infection as mentioned in the discussion (lines 471-474).

Reviewer #2 (Remarks to the Author):

Reviewer #3 (Remarks to the Author):

Dear Authors, The manuscript unravels the differences in the behaviour of EBOV and MARV in Jamaican fruit bats at both the animal and cellular levels, highlighting in particular the specific relationship between certain filoviruses and specific bat species. The data are of great importance to the scientific community and shed light on the interactions between filoviruses and reservoir species. A few points can be raised to reinforce the message of the articles:

Lane 128: Please indicate the history of virus passage.

We have added information on the virus passage history to the methods section (lines 495-500).

Lane 158 : Please provide information on the housing conditions of the bats (surface, diet, anaesthesia).

Information on the cage surface (line 531), diet (line 531), and isoflurane percentage (line 538) was added to the methods.

Lane 304 : The reference refers to another reference.

The reference has been updated to reference the original Takada et al. 1997 manuscript (line 677).

Lane 380 / Suppl Figure 1 / Figure 4L : please add number of events and % in Suppl Figure 1E and add values for all animals as the plot of S1E is currently not very convincing.

The sample size has been added to the figure legend for 4L lines 1069-1072. The raw data with the percentages from Figure 4L using the gating strategy represented in Supplementary Figure 1E are deposited in figshare.

Lane 465 : RNA values are very close to infectious titers, which is not expected for mononegaviruses.

We thank the reviewer for pointing this out. After looking at the titration data, we noticed an error in the TCID₅₀/mL calculation that increased the titer by 5-fold. We have corrected the error and adjusted the limit of detection to 0.5 TCID₅₀/mL.

Lane 478 : Please indicate the viral RNA level in the sera of the animals.

All serum samples were negative by RT-qPCR for viral RNA (Figure 3E and 3F, lines 142-143). We think there is a narrow window of viremia that we have not captured with the evaluated time points.

Lane 488 : Have the adrenal glands been titered for viral RNA and virus, as filoviruses are usually abundant there in susceptible hosts?

The adrenal glands were not specifically collected for the quantification of viral RNA or for virus isolation.

Lane 490 : EBOV and MARV antigen staining would be a real plus, as would caspase 3 staining of spleen sections, as this is a hallmark of filovirus infection in susceptible hosts.

Due to technical difficulties with the EBOV and MARV antibodies and RNA scope, our histology core has not been able to conclusively detect viral antigen/RNA without significant background staining that precludes accurate interpretation. We are actively working to improve the staining protocols in the Jamaican fruit bats, but for now this is not something we can include feasibly. Below are the images provided by the veterinary pathologist.

Negative control skin slide for EBOV RNA scope (200X) show a couple positive spots which prevented us from feeling comfortable with including the data.

Skin of EBOV-infected bat (200x). With more extensive positivity via RNA scope than the negative control.

Spleen (400x) of an EBOV infected bat with RNA scope (left) and IHC (right) with RT-qPCR 6.76 log₁₀ copies/mL, 3.5 log₁₀ TCID₅₀/mL. The pathologist concluded that there is a lot of background precipitate with IHC that is not co-staining on ISH, making the IHC assay questionable.

Lane 506 : as reservoir species are supposed to control but allow viral replication, a check for regulating/proviral ISGs would be of interest (SOCS, USP18, ...).

We agree that understanding how differences in the function ISGs in the negative regulation of innate antiviral pathways could influence the balance of antiviral/pro-inflammatory pathways and have added a point in the discussion (lines 422-424). We

expect that same induction for both antiviral and regulatory ISGs as the transcript's expression is promoter driven.

Lane 540 : Anti-GP antibodies were evaluated to signify the immunisation of the animal, but antibodies raised against nucleoproteins are usually more abundant and may have provided a better signature.

Antibodies to filovirus GP and neutralizing titer were assessed to address the generation of a specific humoral response to EBOV or MARV infection. We agree that NP could have been evaluated as well, but we do not currently have the resources for a filovirus NP ELISA. Given the negativity of the MARV-infected bats for neutralizing titer, except the one that also was seropositive for MARV GP binding antibodies, we do not expect a different result using NP. Further, results Kessler et al. show comparable antibody titers for hemagglutinin, neuraminidase, and nucleoprotein in Jamaican fruit bats infected with H18N11 bat influenza A virus [1].

1. Kessler, S., Stegen, P., Zhan, S., Schwemmler, M., Reuther, P., Schountz, T., Ciminski, K., *Jamaican fruit bats mount a stable and highly neutralizing antibody response after bat influenza virus infection*. Proc Natl Acad Sci U S A, 2024. 121(42): p. e2413619121.

Lane 543 : typo animal EBO420, is it EBO06?

Yes, the text has been updated to EBO06.

Lane 528 : There is more than a one log difference in viral RNA quantity between EBOV and MARV, so the interpretation of transcriptomic data should take this into account.

We have added a sentence (lines 193-196) that state that differences in the transcriptomes are not surprising considering the difference in liver viral load.

Lane 611 / Figure 6E, please comment on the global increase in total STAT.

We believe the reviewer may be referring to Figure 6B, since figure 6E is referring to transcripts of pro-inflammatory cytokines. In lines 269-272, we have mentioned that MARV infection induced the expression of interferon stimulated genes, ISGs, RIG-I and STAT1. In the Figure 6B western blot, there is only an increase in STAT1 and STAT2 protein in the later time points of MARV infected cells, so we are unsure what the reviewer suggests as a global increase.

Lane 670 : Why were Huh7 stimulated?

As stated in lines 333-334, Huh7 cells were selected due to their lack of IFN β production thus allowing the experiment to proceed without an IFN-I induction blockade. Further, these cells are IFN-I responsive and were added a line stating that Huh-7 cells were used to demonstrate MARV VP40 antagonism of IFN-I signaling(lines 334-335).

Lane 748 : Please comment on the hypothermia observed?

Within the results section, we describe the mild hypothermia lines 103-105 and is represented in Figure 1B. In discussion section, we added a sentence (lines 410-413) stating that we cannot rule out the presumed hypothermia could possibly be due to migration of the transponder, but the decrease in body temperature and return to baseline suggests a physiological response to EBOV infection.